# Co-Adaptation of Algorithmic and Implementational Innovations in Inference-based Deep Reinforcement Learning

**Hiroki Furuta**
The University of Tokyo
furuta@weblab.t.u-tokyo.ac.jp

**Tadashi Kozuno**
University of Alberta

**Tatsuya Matsushima**
The University of Tokyo

**Yutaka Matsuo**
The University of Tokyo

**Shixiang Shane Gu**
Google Research

## Abstract

Recently many algorithms were devised for reinforcement learning (RL) with function approximation. While they have clear algorithmic distinctions, they also have many implementation differences that are algorithm-independent and sometimes under-emphasized. Such mixing of algorithmic novelty and implementation craftsmanship makes rigorous analyses of the sources of performance improvements across algorithms difficult. In this work, we focus on a series of off-policy inference-based actor-critic algorithms – MPO, AWR, and SAC – to decouple their algorithmic innovations and implementation decisions. We present unified derivations through a single control-as-inference objective, where we can categorize each algorithm as based on either Expectation-Maximization (EM) or direct Kullback-Leibler (KL) divergence minimization and treat the rest of specifications as implementation details. We performed extensive ablation studies, and identified substantial performance drops whenever implementation details are *mismatched* for algorithmic choices. These results show which implementation or code details are co-adapted and co-evolved with algorithms, and which are transferable across algorithms: as examples, we identified that tanh Gaussian policy and network sizes are highly adapted to algorithmic types, while layer normalization and ELU are critical for MPO's performances but also transfer to noticeable gains in SAC. We hope our work can inspire future work to further demystify sources of performance improvements across multiple algorithms and allow researchers to build on one another's both algorithmic and implementational innovations.[1]

## 1 Introduction

Deep reinforcement learning (RL) has achieved huge empirical successes in both continuous [36, 18] and discrete [39, 26] problem settings with on-policy [52, 54] and off-policy [13, 22, 23] algorithms. Especially in the continuous control domain, interpreting RL as probabilistic inference [60, 61] has yielded many kinds of algorithms with strong empirical performances [34, 50, 51, 12, 29, 21, 16, 24].

Recently, there has been a series of off-policy algorithms derived from this perspective for learning policies with function approximations [2, 22, 48]. Notably, Soft Actor Critic (SAC) [22, 23], based on a maximum entropy objective and soft Q-function, significantly outperforms on-policy [52, 54] and off-policy [36, 18, 13] methods. Maximum a posteriori Policy Optimisation (MPO) [2],

---

[1]The implementation is available at https://github.com/frt03/inference-based-rl.

inspired by REPS [50], employs a pseudo-likelihood objective, and achieves high sample-efficiency and fast convergence compared to the variety of policy gradient methods [36, 54, 7]. Similar to MPO, Advantage Weighted Regression (AWR) [48], and its variant, Advantage Weighted Actor-Critic (AWAC) [42], also employ the pseudo-likelihood objective weighted by the exponential of advantages, and reports more stable performance than baselines both in online and offline [35] settings.

While these inference-based algorithms have similar derivations to each other, their empirical performances have large gaps among them when evaluated on the standard continuous control benchmarks, such as OpenAI Gym [6] or DeepMind Control Suite [58]. Critically, as each algorithm has unique low-level implementation or code design decisions – such as value estimation techniques, action distribution for the policy, and network architectures – aside from high-level algorithmic choices, it is difficult to exactly identify the causes of these performance gaps as algorithmic or implementational.

In this paper, we first derive MPO, AWR, and SAC from a single objective function, through either Expectation-Maximization (EM) or KL minimization, mathematically clarifying the algorithmic connections among the recent state-of-the-art off-policy actor-critic algorithms. This unified derivation allows us to precisely identify implementation techniques and code details for each algorithm, which are residual design choices in each method that are generally transferable to other algorithms. To reveal the sources of the performance gaps, we experiment with carefully-selected ablations of these identified implementation techniques and code details, such as tanh-squashed Gaussian policy, clipped double Q-learning [13], and network architectures. Specifically, we keep the high-level algorithmic designs while normalizing the implementation designs, enabling proper algorithmic comparisons. Our empirical results successfully distinguish between highly co-adapted design choices and no-co-adapted ones[2]. We identified that clipped double Q-learning and tanh-squashed policies, the sources of SoTA performance of SAC, are highly co-adapted, specific to KL-minimization-based method, SAC, and difficult to transfer and benefit in EM-based methods, MPO or AWR. In contrast, we discover that ELU [8] and layer normalization [4], the sources of SoTA performance of MPO, are transferable choices from MPO that also significantly benefit SAC. We hope our work can inspire more future works to precisely decouple algorithmic innovations from implementation or code details, which allows exact sources of performance gains to be identified and algorithmic researches to better build on one another.

## 2 Related Work

**Inference-based RL algorithms** RL as probabilistic inference has been studied in several prior contexts [60, 61, 34, 11, 57, 45], but many recently-proposed algorithms [2, 22, 48] are derived separately and their exact relationships are difficult to get out directly, due to mixing of algorithmic and implementational details, inconsistent implementation choices, environment-specific tunings, and benchmark differences. Our work organizes them as a unified policy iteration method, to clarify their exact mathematical algorithmic connections and tease out subtle, but important, implementation differences. We center our analyses around MPO, AWR, and SAC, because they are representative algorithms that span both EM-based [49, 50, 43, 44, 1, 56, 42] and KL-control-based [59, 51, 12, 29, 21, 33, 32] RL and achieve some of the most competitive performances on popular benchmarks [6, 58]. REPS [50], an EM approach, inspired MPO, AWR, and our unified objective in Eq. 1, while Soft Q-learning [21], a practical extension of KL control to continuous action space through Liu and Wang [38], directly led to the development of SAC.

**Meta analyses of RL algorithms** While many papers propose novel algorithms, some recent works focused on meta analyses of some of the popular algorithms, which attracted significant attention due to these algorithms' high-variance evaluation performances, reproducibility difficulty [9, 28, 65], and frequent code-level optimizations [25, 63, 10, 3]. Henderson et al. [25] empirically showed how these RL algorithms have inconsistent results across different official implementations and high variances even across runs with the same hyper-parameters, and recommended a concrete action item for the community – use more random seeds. Tucker et al. [63] show that high performances of action-dependent baselines [19, 17, 37, 20, 67] were more directly due to different subtle implementation choices. Engstrom et al. [10] focus solely on PPO and TRPO, two on-policy algorithms, and discuss

---

[2]We here regard as *co-adaptation* the indispensable implementation or code decisions that do not stem directly from the conceptual algorithmic development, but from empirical considerations.

| Method | Algorithm | Implementation | | | | |
|--------|-----------|----------------|----------------|----------------|----------------|----------------|
| | | $\pi_q$ update | $\pi_p$ update | $\mathcal{G}$ | $\mathcal{G}$ estimate | $\pi_\theta$ |
| MPO | EM | Analytic + TR | SG + TR | $Q^{\pi_p}$ | Retrace(1) | $\pi_p = \mathcal{N}(\mu_\theta(s), \Sigma_\theta(s))$ |
| AWR | EM | Analytic | Mixture + SG | $A^{\pi_p}$ | TD($\lambda$) | $\pi_p = \mathcal{N}(\mu_\theta(s), \Sigma)$ |
| AWAC | EM | Analytic | Mixture + SG | $Q^{\pi_p}$ | TD(0) | $\pi_p = \mathcal{N}(\mu_\theta(s), \Sigma_\theta)$ |
| SAC | KL | SG | (Fixed to Unif.) | $Q^{\pi_q}_{\text{soft}}$ | TD(0) + TD3 | $\pi_q = \text{Tanh}(\mathcal{N}(\mu_\theta(s), \Sigma_\theta(s)))$ |
| PoWER | EM | Analytic | Analytic | $\eta \log Q^{\pi_p}$ | TD(1) | $\pi_p = \mathcal{N}(\mu_\theta(s), \Sigma_\theta(s))$ |
| RWR | EM | Analytic | SG | $\eta \log r$ | – | $\pi_p = \mathcal{N}(\mu_\theta(s), \Sigma)$ |
| REPS | EM | Analytic | $\pi_q$ | $A^{\pi_p}$ | TD(0) | $\pi_p = \text{Softmax}$ |
| UREX | EM | Analytic | SG | $Q^{\pi_p}$ | TD(1) | $\pi_p = \text{Softmax}$ |
| V-MPO | EM | Analytic + TR | SG + TR | $A^{\pi_p}$ | $n$-step TD | $\pi_p = \mathcal{N}(\mu_\theta(s), \Sigma_\theta(s))$ |
| TRPO | KL | TR | $\pi_q$ | $A^{\pi_p}$ | TD(1) | $\pi_q = \mathcal{N}(\mu_\theta(s), \Sigma_\theta)$ |
| PPO | KL | SG + TR | $\pi_q$ | $A^{\pi_p}$ | GAE | $\pi_q = \mathcal{N}(\mu_\theta(s), \Sigma_\theta)$ |
| DDPG* | KL | SG | (Fixed) | $Q^{\pi_q}$ | TD(0) | $\pi_q = \mu_\theta(s)$ |
| TD3* | KL | SG | (Fixed) | $Q^{\pi_q}$ | TD(0) + TD3 | $\pi_q = \mu_\theta(s)$ |

Table 1: Taxonomy based on the components of inference-based off-policy algorithms: MPO [2], AWR [48], AWAC [42], SAC [22], and other algorithms (see Appendix B). We follow the notation of Sec. 3 & 4. We characterize them with the algorithm family (EM or KL), how policies are updated ($\pi_q$ and $\pi_p$; SG stands for stochastic-gradient-based, and TR stands for trust-region-based), the choice of $\mathcal{G}(\cdot)$, how $\mathcal{G}$ is estimated, and the parameterization of the policy. While the advantage function just can be interpreted as Q-function with baseline subtraction, we explicitly write $A^\pi$ when the state-value function is parameterized, not Q-function. (*Note that DDPG and TD3 are not "inference-based", but can be classified as KL control variants.)

how code-level optimizations, instead of the claimed algorithmic differences, actually led more to PPO's superior performances. Andrychowicz et al. [3] describes low-level (e.g. hyper-parameter choice, and regularization) and high-level (e.g. policy loss) design choices in on-policy algorithms affects the performance of PPO by showing results of large-scale evaluations.

In contrast to those prior works that mainly focus on a single family of on-policy algorithms, PPO and TRPO, and evaluating their implementation details alone, our work focuses on two distinct families of off-policy algorithms, and more importantly, presents unifying mathematical connections among independently-proposed state-of-the-art algorithms. Our experiments demonstrate how *some* implementation choices in Table 1 and code details are co-evolved with algorithmic innovations and/or have non-trivial effects on the performance of off-policy inference-based methods.

## 3 Preliminaries

We consider a Markov Decision Process (MDP) defined by state space $\mathcal{S}$, action space $\mathcal{A}$, state transition probability function $p : \mathcal{S} \times \mathcal{A} \times \mathcal{S} \to [0, \infty)$, initial state distribution $p_1 : \mathcal{S} \to [0, \infty)$, reward function $r : \mathcal{S} \times \mathcal{A} \to \mathbb{R}$, and discount factor $\gamma \in [0, 1)$. Let $R_t$ denote a discounted return $\sum_{u=t}^{\infty} \gamma^{u-t} r(s_u, a_u)$. We assume the standard RL setting, where the agent chooses actions based on a parametric policy $\pi_\theta$ and seeks for parameters that maximize the expected return $\mathbb{E}_{\pi_\theta}[R_1]$. Value functions for a policy $\pi$ are the expected return conditioned by a state-action pair or a state, that is, $Q^\pi(s_t, a_t) := \mathbb{E}_\pi[R_t|s_t, a_t]$, and $V^\pi(s_t) := \mathbb{E}_\pi[R_t|s_t]$. They are called the state-action-value function (Q-function), and state-value function, respectively. The advantage function [5] is defined as $A^\pi(s, a) := Q^\pi(s, a) - V^\pi(s)$. The Q-function for a policy $\pi$ is the unique fixed point of the Bellman operator $\mathcal{T}^\pi$ defined by $\mathcal{T}^\pi Q(s, a) = r(s, a) + \int \pi(a'|s') p(s'|s, a) Q(s', a') \, ds' da'$ for any function $Q : \mathcal{S} \times \mathcal{A} \to \mathbb{R}$. We denote a trajectory or successive state-action sequence as $\tau := (s_1, a_1, s_2, a_2, \dots)$. We also define an unnormalized state distribution under the policy $\pi$ by $d_\pi(s) = \sum_{t=1}^{\infty} \gamma^t p(s_t = s|\pi)$.

**Inference-based Methods** For simplicity, we consider a finite-horizon MDP with a time horizon $T$ for the time being. As a result, a trajectory $\tau$ becomes a finite length: $\tau := (s_1, a_1, \dots, s_{T-1}, a_{T-1}, s_T)$.

As in previous works motivated by probabilistic inference [34, 2], we introduce to the standard graphical model of an MDP a binary event variable $\mathcal{O}_t \in \{0, 1\}$, which represents whether the action in time step $t$ is *optimal* or not. To derive the RL objective, we consider the marginal log-likelihood $\log \Pr(\mathcal{O} = 1|\pi_p)$, where $\pi_p$ is a policy. Note that $\mathcal{O} = 1$ means $\mathcal{O}_t = 1$ for every time step. As is

well known, we can decompose this using a variational distribution $q$ of a trajectory as follows:

$$\log \Pr\left(\mathcal{O} = 1 | \pi_p\right) = \mathbb{E}_q \left[ \log \Pr(\mathcal{O} = 1 | \tau) - \log \frac{q(\tau)}{p(\tau)} + \log \frac{q(\tau)}{p(\tau | \mathcal{O} = 1)} \right]$$
$$= \mathcal{J}(p, q) + D_{KL}(q(\tau) \,||\, p(\tau | \mathcal{O} = 1)),$$

where $\mathcal{J}(p, q) := \mathbb{E}_q \left[ \log \Pr(\mathcal{O} = 1 | \tau) \right] - D_{KL}(q(\tau) \,||\, p(\tau))$ is the evidence lower bound (ELBO). Inference-based methods aim to find the parametric policy which maximizes the ELBO $\mathcal{J}(p, q)$.

There are several algorithmic design choices for $q$ and $\Pr(\mathcal{O} = 1 | \tau)$. Since any $q$ is valid, a popular choice is the one that factorizes in the same way as $p$, that is,

$$p(\tau) = p(s_1) \prod_t p(s_{t+1} | s_t, a_t) \pi_p(a_t | s_t), \qquad q(\tau) = p(s_1) \prod_t p(s_{t+1} | s_t, a_t) \pi_q(a_t | s_t),$$

where $\pi_p$ is a prior policy, and $\pi_q$ is a variational posterior policy. We may also choose which policy ($\pi_p$ or $\pi_q$) to parameterize. As for $\Pr(\mathcal{O} = 1 | \tau)$, the most popular choice is the following one [21, 22, 2, 34, 48]:

$$\Pr(\mathcal{O} = 1 | \tau) \propto \exp\left( \sum_{t=1}^T \eta^{-1} \mathcal{G}(s_t, a_t) \right),$$

where $\eta > 0$ is a temperature, and $\mathcal{G}$ is a function over $\mathcal{S} \times \mathcal{A}$, such as an immediate reward function $r$, Q-function $Q^\pi$, and advantage function $A^\pi$. While we employ $\exp(\cdot)$ in the present paper, there are alternatives [46, 47, 55, 66]: for example, Siegel et al. [55] and Wang et al. [66] consider an indicator function $f : x \in \mathbb{R} \mapsto \mathbb{1}[x \geq 0]$, whereas Oh et al. [46] consider the rectified linear unit $f : x \in \mathbb{R} \mapsto \max\{x, 0\}$.

Incorporating these design choices, we can rewrite the ELBO $\mathcal{J}(p, q)$ in a more explicit form as;

$$\mathcal{J}(\pi_p, \pi_q) = \sum_{t=1}^T \mathbb{E}_q \left[ \eta^{-1} \mathcal{G}(s_t, a_t) - D_{KL}(\pi_q(\cdot | s_t) \,||\, \pi_p(\cdot | s_t)) \right].$$

In the following section, we adopt this ELBO and consider its relaxation to the infinite-horizon setting. Then, starting from it, we derive MPO, AWR, and SAC.

## 4   A Unified View of Inference-based Off-Policy Actor-Critic Algorithms

In Sec. 3, we provided the explicit form of the ELBO $\mathcal{J}(\pi_p, \pi_q)$. However, in practice, it is difficult to maximize it as the expectation $\mathbb{E}_q$ depends on $\pi_q$. Furthermore, since we are interested in a finite-horizon setting, we replace $\sum_{t \in [T]} \mathbb{E}_q$ with $\mathbb{E}_{d_\pi(s)}$. Note that the latter expectation $\mathbb{E}_{d_\pi(s)}$ is taken with the unnormalized state distribution $d_\pi$ under an arbitrary policy $\pi$. This is commonly assumed in previous works [2, 22]. This relaxation leads to the following optimization:

$$\max_{\pi_p, \pi_q} \mathcal{J}(\pi_p, \pi_q) \text{ s.t. } \int d_\pi(s) \int \pi_p(a|s) \, da \, ds = 1 \text{ and } \int d_\pi(s) \int \pi_q(a|s) \, da \, ds = 1, \quad (1)$$

where $\mathcal{J}(\pi_p, \pi_q) = \mathbb{E}_{d_\pi(s)}[\eta^{-1} \mathcal{G}(s, a) - D_{KL}(\pi_q \,||\, \pi_p)]$. With this objective, we can regard recent popular SoTA off-policy algorithms, MPO [2], AWR [48], and SAC [22] as variants of a unified policy iteration method. The components are summarized in Table 1. We first explain how these algorithms can be grouped into two categories of approaches for solving Eq. 1, and then we elaborate additional implementation details each algorithm makes.

### 4.1   Unified Policy Iteration: Algorithmic Perspective

Eq. 1 allows the following algorithmic choices: how or if to parameterize $\pi_p$ and $\pi_q$; what optimizer to use for them; and if to optimize them jointly, or individually while holding the other fixed. We show that the algorithms in Table 1 can be classified into two categories: Expectation-Maximization control (EM control) and direct Kullback-Leibler divergence minimization control (KL control).

### 4.1.1 Expectation-Maximization (EM) Control

This category subsumes MPO [2] (similarly REPS [50]), AWR [48], and AWAC [42], and we term it *EM control*. At high level, the algorithm non-parametrically solves for the variational posterior $\pi_q$ while holding the parametric prior $\pi_p = \pi_{\theta_p}$ fixed (E-Step), and then optimize $\pi_p$ holding the new $\pi_q$ fixed (M-Step). This can be viewed as either a generic EM algorithm and as performing coordinate ascent on Eq. 1. We denote $\theta_p$ and $\pi_q$ *after* iteration $k$ of EM steps by $\theta_p^{(k)}$ and $\pi_q^{(k)}$, respectively.

In **E-step** at iteration $k$, we force the variational posterior policy $\pi_q^{(k)}$ to be close to the optimal posterior policy, i.e., the maximizer of the ELBO $\mathcal{J}(\pi_{\theta_p^{(k-1)}}, \pi_q)$ with respect to $\pi_q$. EM control converts the hard-constraint optimization problem in Eq. 1 to solving the following Lagrangian,

$$
\mathcal{J}(\pi_q, \beta) = \int d_\pi(s) \int \pi_q(a|s) \eta^{-1} \mathcal{G}(s,a) \, dads
$$
$$
- \int d_\pi(s) \int \pi_q(a|s) \log \frac{\pi_q(a|s)}{\pi_{\theta_p^{(k-1)}}(a|s)} \, dads + \beta \left( 1 - \int d_\pi(s) \int \pi_q(a|s) \, dads \right). \quad (2)
$$

We analytically obtain the solution of Eq. 2,

$$
\pi_q^{(k)}(a|s) = Z(s)^{-1} \pi_{\theta_p^{(k-1)}}(a|s) \exp \left( \eta^{-1} \mathcal{G}(s,a) \right),
$$

where $Z(s)$ is the partition function.

In **M-Step** at iteration $k$, we maximize the ELBO $\mathcal{J}(\pi_q^{(k)}, \pi_p)$ with respect to $\pi_p$. Considering the optimization with respect to $\pi_p$ in Eq. 1 results in *forward* KL minimization, which is often referred to as a pseudo-likelihood or policy projection objective,

$$
\max_{\theta_p} \mathbb{E}_{d_\pi(s)\pi_q^{(k)}(a|s)} \left[ \log \pi_{\theta_p}(a|s) \right] = \max_{\theta_p} \mathbb{E}_{d_\pi(s)\pi_{\theta_p^{(k-1)}}(a|s)} \left[ \frac{\log \pi_{\theta_p}(a|s)}{Z(s)} \exp \left( \eta^{-1} \mathcal{G}(s,a) \right) \right],
$$
$$
(3)
$$

where we may approximate $Z(s) \approx \frac{1}{M} \sum_{j=1}^{M} \exp(\eta^{-1} \mathcal{G}(s,a_j))$ with $a_j \sim \pi_{\theta_p^{(k-1)}}(\cdot|s)$ in practice.

### 4.1.2 Direct Kullback-Leibler (KL) Divergence Minimization Control

In contrast to the EM control in Sec. 4.1.1, we only optimize the variational posterior $\pi_q$ while holding the prior $\pi_p$ fixed. In this scheme, $\pi_q$ is parameterized, so we denote it as $\pi_{\theta_q}$. This leads to *KL control* [59, 62, 51, 12, 29, 21].

Equivalent to E-step in Sec. 4.1.1, we force the variational posterior policy $\pi_{\theta_q}$ to be close to the optimal posterior policy, i.e, the maximizer of $\mathcal{J}(\pi_p, \pi_q)$ with respect to $\pi_q$. The difference is that instead of analytically solving $\mathcal{J}(\pi_p, \pi_q)$ for $\pi_q$, we optimize $\mathcal{J}(\pi_p, \pi_{\theta_q})$ with respect to $\theta_q$, which results in the following objective,

$$
\max_{\theta_q} \mathbb{E}_{d_\pi(s)\pi_{\theta_q}(a|s)} \left[ \eta^{-1} \mathcal{G}(s,a) - \log \frac{\pi_{\theta_q}(a|s)}{\pi_p(a|s)} \right].
$$

### 4.1.3 "Optimal" Policies of EM and KL Control

While we formulate EM and KL control in a unified framework, we note that they have a fundamental difference in their definition of "optimal" policies. KL control fixes $\pi_p$ and converges to a *regularized-optimal* policy in an exact case [15, 64]. In contrast, EM control continues updating both $\pi_p$ and $\pi_q$, resulting in convergence to the *standard optimal* policy in an exact case [51]. We also note that EM control solves KL control as a sub-problem; for example, the first E-step exactly corresponds to a KL control problem, except for the policy parameterization.

## 4.2 Unified Policy Iteration: Implementation Details

In this section, we explain the missing pieces of MPO, AWR, and SAC in Sec. 4.1. Additionally, we also describe the details of other algorithms [30, 49, 50, 41, 56, 36, 13, 52, 54] from the EM and KL control perspective. See Appendix B for the details.

### 4.2.1 MPO

**Algorithm**  MPO closely follows the EM control scheme explained in Sec. 4.1.1, wherein $\pi_p = \pi_{\theta_p}$ is parametric, and $\pi_q$ is non-parametric.

**Implementation [$\pi_q$ Update]**  This corresponds to the E-step. MPO uses a trust-region (TR) method. Concretely, it replaces the reverse KL penalty (second term) in the Lagrangian (Eq. 2) with a constraint and analytically solves it for $\pi_q$. As a result, $\eta$ is also optimized during the training by minimizing the dual of Eq. 2, which resembles REPS [50]:

$$g(\eta) = \eta\epsilon + \eta \log \mathbb{E}_{d_\pi(s)\pi_{\theta_p^{(k-1)}}(a|s)} \left[ \exp\left( \eta^{-1} Q^{\pi_{\theta_p^{(k-1)}}}(s,a) \right) \right].$$

**[$\pi_p$ Update]**  This corresponds to the M-step. MPO uses a combination of Stochastic Gradient (SG) ascent and trust-region method based on a forward KL divergence similarly to TRPO [52]. Concretely, it obtains $\theta_p^{(k)}$ by maximizing the objective (Eq. 3) with respect to $\theta_p$ subject to $\mathbb{E}_{d_\pi(s)}[D_{KL}(\pi_{\theta_p^{(k-1)}}(\cdot|s) \,||\, \pi_{\theta_p}(\cdot|s))] \leq \varepsilon$. MPO further decompose this KL divergence to two terms, assuming Gaussian policies; one term includes only the mean vector of $\pi_{\theta_p}(\cdot|s)$, and the other includes only its covariance matrix. Abdolmaleki et al. [2] justify this as a log-prior of MAP estimation, which is assumed as a second-order approximation of KL divergence.

**[$\mathcal{G}$ and $\mathcal{G}$ Estimate]**  MPO uses the Q-function of $\pi_{\theta_p^{(k-1)}}$ as $\mathcal{G}$ in the $k$-th E-step. It originally uses the Retrace update [40], while its practical implementation [27] uses a single-step Bellman update.

**[$\pi_\theta$]**  For a parameterized policy, MPO uses a Gaussian distribution with state-dependent mean vector and diagonal covariance matrix. The trust-region method in MPO's M-step heavily relies on this Gaussian assumption. Since a Gaussian distribution has an infinite support, MPO has a penalty term in its policy loss function that forces the mean of the policy to stay within the range of action space.

### 4.2.2 AWR and AWAC

**Algorithm**  AWR slightly deviates from the EM control. In the M-step, AWR simply set $Z(s)$ to 1.

**Implementation [$\pi_q$ Update]**  This corresponds the E-step. AWR and AWAC analytically solves the Lagrangian (Eq. 2). In contrast to MPO, they don't use trust-region method.

**[$\pi_p$ Update]**  This corresponds to the M-step. At iteration $k$, AWR uses an average of all previous policies $\widetilde{\pi}_{p^k} := \frac{1}{k}\sum_{j=0}^{k-1} \pi_{\theta_p^{(j)}}$ instead of $\pi_{\theta_p^{(k-1)}}$ (cf. Eq. 3). In practice, the average policy $\widetilde{\pi}_{p^k}$ is replaced by samples of actions from a replay buffer, which stores action samples of previous policies.

**[$\mathcal{G}$ and $\mathcal{G}$ Estimate]**  AWR uses the advantage function of $\widetilde{\pi}_{p^k}$ as $\mathcal{G}$ in the $k$-th E-step, and learns the state-value function of $\pi_{\theta_p^{(k-1)}}$ with TD($\lambda$) [53]. Due to its choice of $\widetilde{\pi}_{p^k}$, this avoids importance corrections [40]. In contrast, AWAC estimates the advantage via the Q-function with TD(0).

**[$\pi_\theta$]**  For a parameterized policy, both AWR and AWAC use a Gaussian distribution with state-dependent mean vector and state-independent diagonal covariance matrix (a constant one for AWR). As in MPO, they uses a penalty term to keep the mean of the policy within the range of action space.

### 4.2.3 SAC

**Algorithm**  Contrary to MPO and AWR, SAC follows the KL control scheme explained in Sec. 4.1.2, wherein the variational posterior policy $\pi_q = \pi_{\theta_q}$ is parameterized, and the prior policy $\pi_p$ is fixed to the uniform distribution over the action space $\mathcal{A}$. SAC uses as $\mathcal{G}$ a soft Q-function:

$$Q_{\text{soft}}^{\pi_{\theta_q}}(s_t, a_t) := r(s_t, a_t) + \gamma \mathbb{E}_{\pi_{\theta_q}}\left[ V_{\text{soft}}^{\pi_{\theta_q}}(s_{t+1}) \right],$$

$$V_{\text{soft}}^{\pi_{\theta_q}}(s_t) := V^{\pi_{\theta_q}}(s_t) + \mathbb{E}_{\pi_{\theta_q}}\left[ \sum_{u=t}^{\infty} \gamma^{u-t} \eta \mathcal{H}(\pi_{\theta_q}(\cdot|s_t)) \big| s_t \right],$$

with $\mathcal{H}(\pi_{\theta_q}(\cdot|s_t))$ being $-\mathbb{E}_{\pi_{\theta_q}}[\log \pi_{\theta_q}(a_t|s_t)|s_t]$.

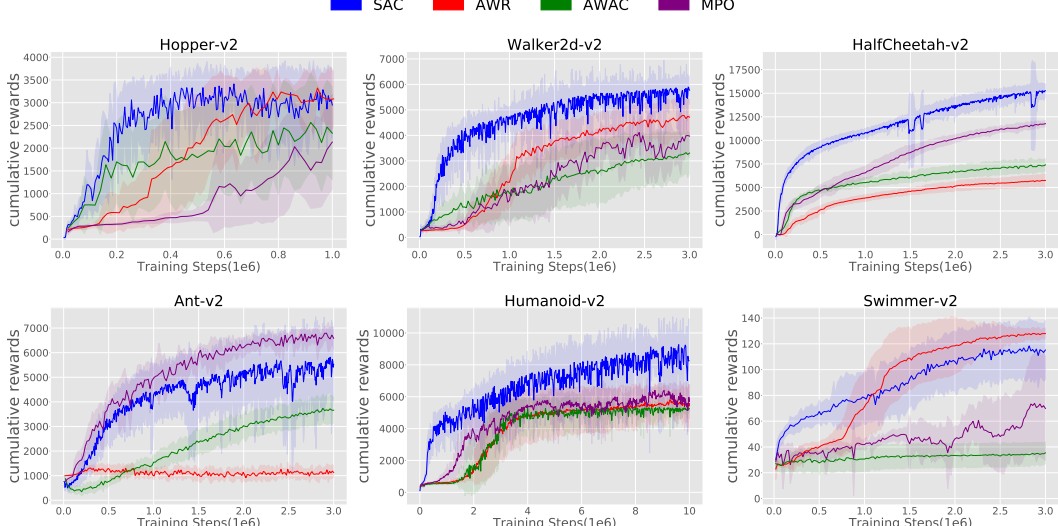

Figure 1: Benchmarking results on OpenAI Gym MuJoCo locomotion environments. All methods are run with 10 random seeds. SAC seems to perform consistently better.

**Implementation [$\pi_q$ and $\pi_p$ Update]** SAC performs stochastic gradient (SG) ascent updates of only $\pi_q$, where the objective function is shown in Eq. 1. SAC has no operation equivalent to M-step in MPO and AWR, since it keeps the prior policy $\pi_p$ to the uniform distribution.

Similar to the temperature tuning in MPO, Haarnoja et al. [23] consider the dual function of entropy-constrained Lagrangian and treat the temperature $\eta$ as a Lagrange multiplier, which results in,

$$g(\eta) = -\eta\bar{\mathcal{H}} - \eta\mathbb{E}_{d_\pi(s)\pi_{\theta_q}(a|s)}\left[\log\pi_{\theta_q}(a|s)\right],$$

where $\bar{\mathcal{H}}$ is the target entropy to ensure that the entropy of the policy should be larger than it.

**[$\mathcal{G}$ and $\mathcal{G}$ Estimate]** SAC uses $Q^{\pi_{\theta_q}}_{\text{soft}}$ as $\mathcal{G}$, and TD(0)-like algorithm based on this soft Q-function. In policy evaluation, clipped double Q-learning [13], which retains multiple Q-functions (typically two) and takes the minimum, is employed to suppress the overestimation of Q value.

**[$\pi_\theta$]** The policy of SAC is a squashed Gaussian distribution: SAC first samples a random variable $u$ from a Gaussian distribution $\mathcal{N}(\mu_\theta(s), \Sigma_\theta(s))$ with state-dependent mean vector and diagonal covariance; then, it applies the tanh function to $u$ and obtain the action $a \in [-1, 1]^{|\mathcal{A}|}$. (If a dimension of the action space is not $[-1, 1]$, an appropriate scaling and shift by an affine function is applied after the squashing.) Tanh squashing prevents out-of-bounds action.

### 4.3 Empirical Comparison

To evaluate the empirical performance of these off-policy algorithms (MPO, AWR, AWAC, and SAC), we compare their performances on Open AI Gym MuJoCo environments, namely, Hopper, Walker2d, HalfCheetah, Ant, Humanoid, and Swimmer, following Haarnoja et al. [23]. We reproduce all algorithms based on pytorch RL library [14], referring their original implementations [22, 48, 42, 27]. Figure 1 shows that SAC outperforms others in four environments (Hopper, Walker2d, HalfCheetah, and Humanoid), while MPO in Ant and AWR in Swimmer achieves the best performance. Generally, SAC seems to perform consistently better. We extensively evaluate MPO, AWR, and SAC on the 28 tasks on DeepMind Control Suite and 3 MuJoCo manipulation tasks. See Appendix C and D for the details.

## 5 Evaluation on Implementational Choices

In Sec. 4.3, SAC shows notable results in most environments, while we revealed that the derivation and formulation of those methods resemble each other. To specify the effect of each implementational

|  | SAC (D) | SAC (S) | AWAC (D) | AWAC (S) | MPO (D) | MPO (S) |
|---|---|---|---|---|---|---|
| Hopper-v2 | **3013 ± 602** | 1601 ± 733 | 2329 ± 1020 | 2540 ± 755 | 2352 ± 959 | 2136 ± 1047 |
| Walker2d-v2 | **5820 ± 411** | 1888 ± 922 | 3307 ± 780 | 3662 ± 712 | 4471 ± 281 | 3972 ± 849 |
| HalfCheetah-v2 | 15254 ± 751 | **15701 ± 630** | 7396 ± 677 | 7226 ± 449 | 12028 ± 191 | 11769 ± 321 |
| Ant-v2 | 5532 ± 1266 | 1163 ± 1326 | 3659 ± 523 | 3008 ± 375 | **7179 ± 190** | 6584 ± 455 |
| Humanoid-v2 | **8081 ± 1149** | 768 ± 215 | 5243 ± 200 | 2738 ± 982 | 6858 ± 373 | 5709 ± 1081 |
| Swimmer-v2 | 114 ± 21 | **143 ± 3** | 35 ± 8 | 38 ± 7 | 69 ± 29 | 70 ± 40 |

Table 2: Ablation of Clipped Double Q-Learning. (D) denotes algorithms with clipped double Q-learning, and (S) denotes without it. We test original SAC (D), AWAC (D), MPO (S) and some variants; SAC without clipped double Q (S), AWAC (S), and MPO with clipped double Q (D). SAC (S) beats SAC (D) in HalfCheetah and Swimmer, while it fails in Hopper, Walker, Ant and Humanoid, which implies that SAC (S) obtains a highly exploratory policy, since it fails in termination environments. The learning curves are shown in Appendix F.

or code detail on the performance, we experiment with extensive and careful one-by-one ablations on; (1) clipped double Q-learning, (2) action distribution for the policy, (3) activation and normalization, and (4) network size. The former two correspond to implementation details (the choice of $\mathcal{G}$ estimate and the parameterization of the policy), and the latter two correspond to code details (see Appendix G for further experiments on other implementation details, such as $\pi_p$ update or the choice of $\mathcal{G}$). We also conclude several recommendations for the practitioners (Table 6).

## 5.1 Clipped Double Q-Learning

We hypothesize that clipped double Q-learning has a large effect on the notable performance of SAC, and could be transferable in EM control methods. To verify this, we test the effect of clipped double Q-learning. Instead of AWR, here we evaluate AWAC since it uses Q-function. Table 2 shows that single-Q SAC outperforms original one in HalfCheetah and Swimmer that do not have the termination of the episode, while struggles to learn in Hopper, Walker, Ant and Humanoid that have the episodic termination conditions. A hypothesis is that single-Q SAC obtains a more exploratory policy due to larger overestimation bias, which can help in environments where explorations are safe, but hurt in environments where reckless explorations lead to immediate terminations. In contrast, clipped double Q-learning does not help MPO or AWAC as significantly as SAC; most results do not change or lead to slight improvements over the originals. This suggests that clipped double Q-learning might be a co-dependent and indispensable choice to KL control methods.

**Recommendation** Use clipped double Q-learning as a default choice, but you can omit it in EM control methods or non-terminal environments, such as HalfCheetah or Swimmer.

## 5.2 Action Distribution for the Policy

Another hypothesis is that the tanh-squashed policy (last column in Table 1) is an important and transferable design choice. We compare SAC without tanh transform (with MPO action penalty instead) to the original one, which results in drastic degradation (Table 3). This can be caused by the maximum entropy objective that encourages maximizing the covariance. These observations suggest that the high performance of SAC seems to highly depend on the implementational choice of the policy distribution. In contrast, MPO and AWR with tanh squashing don't enjoy such large performance gain, or achieve worse cumulative rewards. This also suggests that tanh-squashed policy might be highly co-adapted in SAC and less transferable to EM control methods. Note that for EM control, we must clip the actions to keep them within the supports of distributions; $a \in [-1 + \epsilon, 1 - \epsilon]^{|\mathcal{A}|}$. We found that no clipping cause significant numerical instability. See Appendix H for the details.

**Recommendation** Use the original distributions for each algorithm. If you use the tanh-squashed Gaussian in EM control methods, clip the action to be careful for the numerical instability.

## 5.3 Activation and Normalization

The implementation of MPO [27] has some code-level detailed choices; ELU activation [8] and layer normalization [4]. We hypothesize that these, sometimes unfamiliar, code details in practical implementation stabilize the learning process, and contribute to the performance of MPO most.

| | SAC (w/) | SAC (w/o) | AWR (w/) | AWR (w/o) | MPO (w/) | MPO (w/o) |
|---|---|---|---|---|---|---|
| Hopper-v2 | $3013 \pm 602$ | $6 \pm 10$ | $2709 \pm 905$ | $\mathbf{3085 \pm 593}$ | $2149 \pm 849$ | $2136 \pm 1047$ |
| Walker2d-v2 | $\mathbf{5820 \pm 411}$ | $-\infty$ | $3295 \pm 335$ | $4717 \pm 678$ | $3167 \pm 815$ | $3972 \pm 849$ |
| HalfCheetah-v2 | $\mathbf{15254 \pm 751}$ | $-\infty$ | $3653 \pm 652$ | $5742 \pm 667$ | $9523 \pm 312$ | $11769 \pm 321$ |
| Ant-v2 | $5532 \pm 1266$ | $-\infty$ | $445 \pm 106$ | $1127 \pm 224$ | $2880 \pm 306$ | $\mathbf{6584 \pm 455}$ |
| Humanoid-v2 | $\mathbf{8081 \pm 1149}$ | $108 \pm 82^\dagger$ | $2304 \pm 1629^\dagger$ | $5573 \pm 1020$ | $6688 \pm 192$ | $5709 \pm 1081$ |
| Swimmer-v2 | $114 \pm 21$ | $28 \pm 11$ | $121 \pm 3$ | $\mathbf{128 \pm 4}$ | $110 \pm 42$ | $70 \pm 40$ |

Table 3: Ablation of Tanh transformation ($^\dagger$numerical error happens during training). We test SAC without tanh squashing, AWR with tanh, and MPO with tanh. SAC without tanh transform results in drastic degradation of the performance, which can be caused by the maximum entropy objective that encourages the maximization of the covariance. In contrast, EM Control methods don't enjoy the performance gain from the tanh-squashed policy, which seems a less transferable choice. The learning curves are shown in Appendix F.

| | Hopper-v2 | Walker2d-v2 | HalfCheetah-v2 | Ant-v2 | Humanoid-v2 | Swimmer-v2 |
|---|---|---|---|---|---|---|
| **SAC** | $3013 \pm 602$ | $\mathbf{5820 \pm 411}$ | $15254 \pm 751$ | $5532 \pm 1266$ | $8081 \pm 1149$ | $114 \pm 21$ |
| **SAC-E$^+$** | $2337 \pm 903$ | $5504 \pm 431$ | $\mathbf{15350 \pm 594}$ | $6457 \pm 828$ | $\mathbf{8196 \pm 892}$ | $\mathbf{146 \pm 7}$ |
| **SAC-L$^+$** | $2368 \pm 179$ | $5613 \pm 762$ | $13074 \pm 2218$ | $\mathbf{7349 \pm 176}$ | $8146 \pm 470$ | $99 \pm 18$ |
| **SAC-E$^+$L$^+$** | $1926 \pm 417$ | $5751 \pm 400$ | $12555 \pm 1259$ | $7017 \pm 132$ | $7687 \pm 1385$ | $143 \pm 9$ |
| **MPO** | $2136 \pm 1047$ | $3972 \pm 849$ | $11769 \pm 321$ | $6584 \pm 455$ | $5709 \pm 1081$ | $70 \pm 40$ |
| **MPO-E$^-$** | $2700 \pm 879$ | $3553 \pm 1145$ | $11638 \pm 664$ | $5917 \pm 702$ | $4870 \pm 1917$ | $108 \pm 28$ |
| **MPO-L$^-$** | $824 \pm 250$ | $2413 \pm 1352$ | $6064 \pm 4596$ | $2135 \pm 2988$ | $5039 \pm 838$ | $-\infty$ |
| **MPO-E$^-$L$^-$** | $843 \pm 168$ | $1708 \pm 663$ | $-1363 \pm 20965$ | $807 \pm 2351$ | $5566 \pm 787$ | $-\infty$ |
| **AWR** | $3085 \pm 593$ | $4717 \pm 678$ | $5742 \pm 667$ | $1127 \pm 224$ | $5573 \pm 1020$ | $128 \pm 4$ |
| **AWR-E$^+$** | $1793 \pm 1305$ | $4418 \pm 319$ | $5910 \pm 754$ | $2288 \pm 715$ | $6708 \pm 226$ | $128 \pm 4$ |
| **AWR-L$^+$** | $2525 \pm 1130$ | $4900 \pm 671$ | $5391 \pm 232$ | $639 \pm 68$ | $5962 \pm 376$ | $129 \pm 2$ |
| **AWR-E$^+$L$^+$** | $\mathbf{3234 \pm 118}$ | $4906 \pm 304$ | $6081 \pm 753$ | $2283 \pm 927$ | $6041 \pm 270$ | $130 \pm 1$ |

Table 4: Incorporating ELU/layer normalization into SAC and AWR. E$^+$/L$^+$ indicates adding, and E$^-$/L$^-$ indicates removing ELU/layer normalization. Introducing layer normalization or ELU into SAC improves the performances in Ant (beating MPO), Swimmer (beating AWR), HalfCheetah, and Humanoid. AWR also shows the improvement in several tasks. MPO removing layer normalization largely drops its performance. Both code-level details seems transferable between EM and KL control methods.

Especially in Ant (Sec. 4.3), MPO significantly outperforms SAC. To investigate this much deeper, we add them into SAC or AWR and remove them from MPO, while maintaining the rest of the implementations. Table 4 shows that layer normalization can contribute to a significantly higher performance of SAC in Ant, and replacing ReLU with ELU also improves performance a lot in Swimmer, where AWR is the best in Figure 1. In contrast, the performance of MPO drastically collapsed when we just removed ELU and layer normalization. This observation suggests that these code-level choices are not only indispensable for MPO, but transferable and beneficial to both KL and EM control methods. Additionally, we tested incorporating ELU and layer normalization to SAC in 12 DM Control tasks where MPO outperformed SAC, and observed that they again often benefit SAC performances substantially. See Appendix C for the details.

**Recommendation**   It is worth considering to replace the activation function from ReLU to ELU and incorporate the layer normalization, to achieve the best performance in several tasks. Both of them are transferable between EM and KL control methods.

### 5.4  Network Size

While on-policy algorithms such as PPO and TRPO can use the common network architecture as in prior works [10, 25], the architecture that works well in all the off-policy inference-based methods is still not obvious, and the RL community does not have agreed upon default choice. To validate the dependency on the networks among the inference-based methods, we exchange the size and number of hidden layers in the policy and value networks. We denote the network size of MPO ((256, 256, 256) for policy and (512, 512, 256) for value) as large (L), of SAC ((256, 256) for policy and value) as middle (M), and of AWR ((128, 64) for policy and value) as small (S) (see Appendix A for the details). Table 5 illustrates that SAC with large network seems to have better performance. However, as shown in Appendix F (Figure 8), the learning curve sometimes becomes unstable. While the certain degree of robustness to the network size is observed in SAC, EM control methods, MPO

|  | Hopper-v2 | Walker2d-v2 | HalfCheetah-v2 | Ant-v2 | Humanoid-v2 | Swimmer-v2 |
|---|---|---|---|---|---|---|
| **SAC (L)** | $2486 \pm 746$ | $3188 \pm 2115$ | $\mathbf{16528 \pm 183}$ | $\mathbf{7495 \pm 405}$ | $\mathbf{8255 \pm 578}$ | $118 \pm 26$ |
| **SAC (M)** | $3013 \pm 602$ | $\mathbf{5820 \pm 411}$ | $15254 \pm 751$ | $5532 \pm 1266$ | $8081 \pm 1149$ | $114 \pm 21$ |
| **SAC (S)** | $\mathbf{3456 \pm 81}$ | $4939 \pm 284$ | $12241 \pm 400$ | $3290 \pm 691$ | $7724 \pm 497$ | $59 \pm 11$ |
| **MPO (L)** | $2136 \pm 1047$ | $3972 \pm 849$ | $11769 \pm 321$ | $6584 \pm 455$ | $5709 \pm 1081$ | $70 \pm 40$ |
| **MPO (M)** | $661 \pm 79$ | $1965 \pm 1426$ | $-\infty$ | $5192 \pm 538$ | $6015 \pm 771$ | $81 \pm 28$ |
| **MPO (S)** | $430 \pm 99$ | $2055 \pm 990$ | $5003 \pm 1567$ | $3587 \pm 957$ | $4745 \pm 1428$ | $59 \pm 28$ |
| **AWR (L)** | $3221 \pm 193$ | $4688 \pm 648$ | $4360 \pm 542$ | $35 \pm 43$ | $665 \pm 54$ | $\mathbf{133 \pm 3}$ |
| **AWR (M)** | $2816 \pm 910$ | $4826 \pm 547$ | $5538 \pm 720$ | $413 \pm 117$ | $3849 \pm 1647$ | $\mathbf{133 \pm 2}$ |
| **AWR (S)** | $3085 \pm 593$ | $4717 \pm 678$ | $5742 \pm 667$ | $1127 \pm 224$ | $5573 \pm 1020$ | $128 \pm 4$ |

Table 5: The performance of each algorithm with different network size. (S) stands for the small network size from AWR, (M) for the middle network size from SAC, and (L) for the large network size from MPO. Generally, SAC, a KL control method seems more robust to the network size than EM control methods; MPO and AWR.

and AWR, seem fragile and more dependent on the specific network size. This trend is remarkable in MPO. AWR (M) or (L) also struggles to learn in high-dimensional state tasks, such as Ant (111 dim), or Humanoid (376 dim). The results also imply that, in contrast to on-policy methods [3], the network size in the off-policy inference-based methods seems a less transferable choice.

**Recommendation** For SAC, use medium size network. Large size will also work, but the learning curve might be unstable. For MPO, we strongly recommend to stick to large size, because it is very sensitive to the network size. For AWR, using small size is a better choice, especially in high-dimensional state tasks, such as Ant, or Humanoid.

# 6 Conclusion

In this work, we present a taxonomy of inference-based algorithms, and successfully identify algorithm-specific as well as algorithm-independent implementation details that cause substantial performance improvements. We first reformulated recent inference-based off-policy algorithms – such as MPO, AWR and SAC – into a unified mathematical objective and exhaustively clarified the algorithmic and implementational differences. Through precise ablation studies, we empirically show that implementation choices like tanh-squashed distribution and clipped double Q-learning are highly co-adapted to KL control methods (e.g. SAC), and difficult to benefit in EM control methods (e.g. MPO or AWR). As an example, the network archi-

|  | MPO | AWR | SAC |
|---|---|---|---|
| Clipped Double Q [5.1] | △ | △ | ◯ |
| Tanh Gaussian [5.2] | △ | △ | ◯ |
| ELU & LayerNorm [5.3] | ◯ | ◯ | ◯ |
| Large Network [5.4] | ◯ | △ | ◯ |
| Medium Network [5.4] | × | △ | ◯ |
| Small Network [5.4] | × | ◯ | △ |

Table 6: Intuitive summary of the ablations. ◯ stands for indispensable choice, ◯ stands for recommended choice, △ stands for not much different or worse choice than expected, and × stands for un-recommended choice. See **Recommendation** for the details.

tectures of inference-based off-policy algorithms, especially EM controls, seem more co-dependent than on-policy methods like PPO or TRPO, which therefore have significant impacts on the overall algorithm performances and need to be carefully tuned per algorithm. Such dependence of each algorithmic innovation on specific hand-tuned implementation details makes accurate performance gain attributions and cumulative build-up of research insights difficult. In contrast, we also find that some code-level implementation details, such as ELU and layer normalization, are not only indispensable choice to MPO, but also transferable and beneficial to SAC substantially. We hope our work can encourage more works that study precisely the impacts of algorithmic properties and empirical design choices, not only for one type of algorithms, but also across a broader spectrum of deep RL algorithms.

**Acknowledgements**

We thank Yusuke Iwasawa, Masahiro Suzuki, Marc G. Bellemare, Ofir Nachum, and Sergey Levine for many fruitful discussions and comments. This work has been supported by the Mohammed bin Salman Center for Future Science and Technology for Saudi-Japan Vision 2030 at The University of Tokyo (MbSC2030).

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
