# Appendix

## A Network Architectures

In this section, we describe the details of the network architectures used in Sec. 4 and 5.

We mainly used 4 GPUs (NVIDIA V100; 16GB) for the experiments in Sec. 4 and 5 and it took about 4 hours per seed (in the case of 3M steps). Actually, we conducted exhaustive evaluations through the enormous experiments, and we hope our empirical observations and recommendations help the practitioners to explore the explosive configuration space.

| Architecture | MPO | AWR | AWAC | SAC |
|---|---|---|---|---|
| Policy network | (256, 256, 256) | (128, 64) | (256, 256) | (256, 256) |
| Value network | (512, 512, 256) | (128, 64) | (256, 256) | (256, 256) |
| Activation function | ELU | ReLU | ReLU | ReLU |
| Layer normalization | ✓ | – | – | – |
| Input normalization | – | ✓ | – | – |
| Optimizer | Adam | SGD (momentum=0.9) | Adam | Adam |
| Learning rate (policy) | 1e-4 | 5e-5 | 3e-4 | 3e-4 |
| Learning rate (value) | 1e-4 | 1e-2 | 3e-4 | 3e-4 |
| Weight initialization | Uniform | Xavier Uniform | Xavier Uniform | Xavier Uniform |
| Initial output scale (policy) | 1.0 | 1e-4 | 1e-2 | 1e-2 |
| Target update | Hard | – | Soft (5e-3) | Soft (5e-3) |
| Clipped Double Q | False | – | True | True |

Table 7: Details of each network architecture. We refer the original implementations of each algorithm which is available online [23, 14, 48, 27, 42]. Note that AWR uses different learning rates of the policy per environment.

| MPO | Hopper-v2 | Walker2d-v2 | HalfCheetah-v2 | Ant-v2 | Humanoid-v2 | Swimmer-v2 |
|---|---|---|---|---|---|---|
| Learning rate ($\eta$) | | | 1e-2 | | | |
| Dual constraint | | | 1e-1 | | | |
| Mean constraint | 3.34e-4 | 1.67e-4 | 1e-3 | 1e-3 | 5.88e-5 | 1e-3 |
| Stddev constraint | 3.34e-7 | 1.67e-7 | 1e-6 | 1e-6 | 5.88e-8 | 1e-6 |
| Action penalty constraint | | | 1e-3 | | | |
| Inital stddev scale | 0.7 | 0.3 | 0.5 | 0.5 | 0.3 | 0.5 |
| Discount factor $\gamma$ | | | 0.99 | | | |

Table 8: Hyper-parameters of MPO. We follow the implementation by Hoffman et al. [27]. Some of mean & stddev constraint are divided by the number of dimensions in the action space as suggested by Hoffman et al. [27], which is empirically better.

| AWR | Hopper-v2 | Walker2d-v2 | HalfCheetah-v2 | Ant-v2 | Humanoid-v2 | Swimmer-v2 |
|---|---|---|---|---|---|---|
| Learning rate (policy) | 1e-4 | 2.5e-5 | 5e-5 | 5e-5 | 1e-5 | 5e-5 |
| Stddev scale | 0.4 | 0.4 | 0.4 | 0.2 | 0.4 | 0.4 |
| Exp-Advantage Weight clip | | | 20.0 | | | |
| Action penalty coefficient | | | 10.0 | | | |
| Discount factor $\gamma$ | | | 0.99 | | | |
| $\lambda$ for TD($\lambda$) | | | 0.95 | | | |

Table 9: Hyper-parameters of AWR. We follow the implementation by Peng et al. [48].

**Small Network (AWR)** We denote the policy and value network used in AWR as a small (S) network, described as follows (in Sec. 5.4, we didn't change the `activation` and `distribution`):

```
from torch import nn

activation = nn.ReLU()
distribution = GaussianHeadWithFixedCovariance()

policy = nn.Sequential(
    nn.Linear(obs_size, 128),
```

```
        activation,
        nn.Linear(128, 64),
        activation,
        nn.Linear(64, action_size),
        distribution,
)
vf = nn.Sequential(
        nn.Linear(obs_size, 128),
        activation,
        nn.Linear(128, 64),
        activation,
        nn.Linear(64, 1),
)
```

**Medium Network (SAC)** We denote the policy and value network used in SAC as a medium (M) network, described as follows (in Sec. 5.4, we didn't change the `activation` and `distribution`):

```
from torch import nn

activation = nn.ReLU()
distribution = TanhSquashedDiagonalGaussian()

policy = nn.Sequential(
        nn.Linear(obs_size, 256),
        activation,
        nn.Linear(256, 256),
        activation,
        nn.Linear(256, action_size * 2),
        distribution
)
q_func = nn.Sequential(
        ConcatObsAndAction(),
        nn.Linear(obs_size + action_size, 256),
        activation,
        nn.Linear(256, 256),
        activation,
        nn.Linear(256, 1)
)
```

**Large Network (MPO)** We denote the policy and value network used in MPO as a large (L) network, described as follows (in Sec. 5.4, we didn't change the `activation` and `distribution`):

```
from torch import nn

activation = nn.ELU()
distribution = GaussianHeadWithDiagonalCovariance()

policy = nn.Sequential(
        nn.Linear(obs_size, 256),
        nn.LayerNorm(256),
        nn.Tanh(),
        activation,
        nn.Linear(256, 256),
        activation,
        nn.Linear(256, 256),
        activation,
        nn.Linear(256, action_size * 2),
        distribution
)
q_func = nn.Sequential(
        ConcatObsAndAction(),
        nn.Linear(obs_size + action_size, 512),
        nn.LayerNorm(512),
        nn.Tanh(),
        activation,
```

```
    nn.Linear(512, 512),
    activation,
    nn.Linear(512, 256),
    activation,
    nn.Linear(256, 1)
)
```

# B    Relations to Other Algorithms

We here explain the relation of the unified policy iteration scheme covers other algorithms. While we mainly focused on AWR, MPO, and SAC in the this paper, our unified scheme covers other algorithms too, as summarized in Table 1:

**EM control algorithms:**

- PoWER [30]: $\pi_p$ $(= \pi_\theta)$ update is analytic. $\mathcal{G} = \eta \log Q^{\pi_p}$ and $Q^{\pi_p}$ is estimated by TD(1). $\pi_\theta = \mathcal{N}(\mu_\theta(s), \Sigma_\theta(s))$.
- RWR [49]: $\pi_p = \pi_\theta$ is updated by SG. $\mathcal{G} = \eta \log r$, and $\pi_\theta = \mathcal{N}(\mu_\theta(s), \Sigma)$. When the reward is unbounded, RWR requires adaptive reward transformation (e.g. $u_\beta(r(s, a)) = \beta \exp(-\beta r(s, a))$; $\beta$ is a learnable parameter).
- REPS [50]: $\pi_p$ is $\pi_q$ of the previous EM step (on-policy) or a mixture of all previous $\pi_q$ (off-policy), which is approximated by samples. $\mathcal{G} = A^{\pi_p}$ and estimated by a single-step TD error with a state-value function computed by solving a dual function. $\pi_q$ is assumed as a softmax policy for discrete control in the original paper.
- UREX [41]: $\pi_p = \pi_\theta$ is updated by SG. $\mathcal{G} = Q^{\pi_p}$ and estimated by TD(1). $\pi_\theta$ is assumed as a softmax policy for discrete control in the original paper.
- V-MPO [56]: Almost the same as MPO, but a state-value function is trained by $n$-step bootstrap instead of Q-function. Top-K advantages are used in E-step.

**KL control algorithms:**

- TRPO [52]: $\pi_q = \pi_\theta = \mathcal{N}(\mu_\theta(s), \Sigma_\theta)$. The KL penalty is converted to a constraint, and the direction of the KL is reversed. $\mathcal{G} = A^{\pi_p}$ and estimated by TD(1). $\pi_p$ is continuously updated to $\pi_q$.
- PPO with a KL penalty [54]: $\pi_q = \pi_\theta = \mathcal{N}(\mu_\theta(s), \Sigma_\theta)$. The direction of the KL penalty is reversed. $\mathcal{G} = A^{\pi_p}$ and estimated by GAE [53]. An adaptive $\eta$ is used so that $D_{KL}(\pi_p \parallel \pi_q)$ approximately matches to a target value.
- DDPG[3] [36]: $\pi_q = \pi_\theta$ is the delta distribution and updated by SG. $\mathcal{G} = Q^{\pi_q}$ and estimated by TD(0). $\eta = 0$ (i.e., the KL divergence and $\pi_p$ update are ignored).
- TD3[2] [13]: It is a variant of DDPG and leverages three implementational techniques, clipped double Q-learning, delayed policy updates, and target policy smoothing.
- BRAC [68] and BEAR [31] (Offline RL): When we assume $\pi_p = \pi_b$ (any behavior policy), and omitting its update, some of the offline RL methods, such as BRAC or BEAR, can be interpreted as one of the KL control methods. Both algorithms utilize the variants of clipped double Q-learning ($\lambda$-interpolation between max and min).

---

[3]Note that DDPG and TD3 are not the "inference-based" algorithms, but we can classify these two as KL control variants.

# C Benchmarks on DeepMind Control Suite

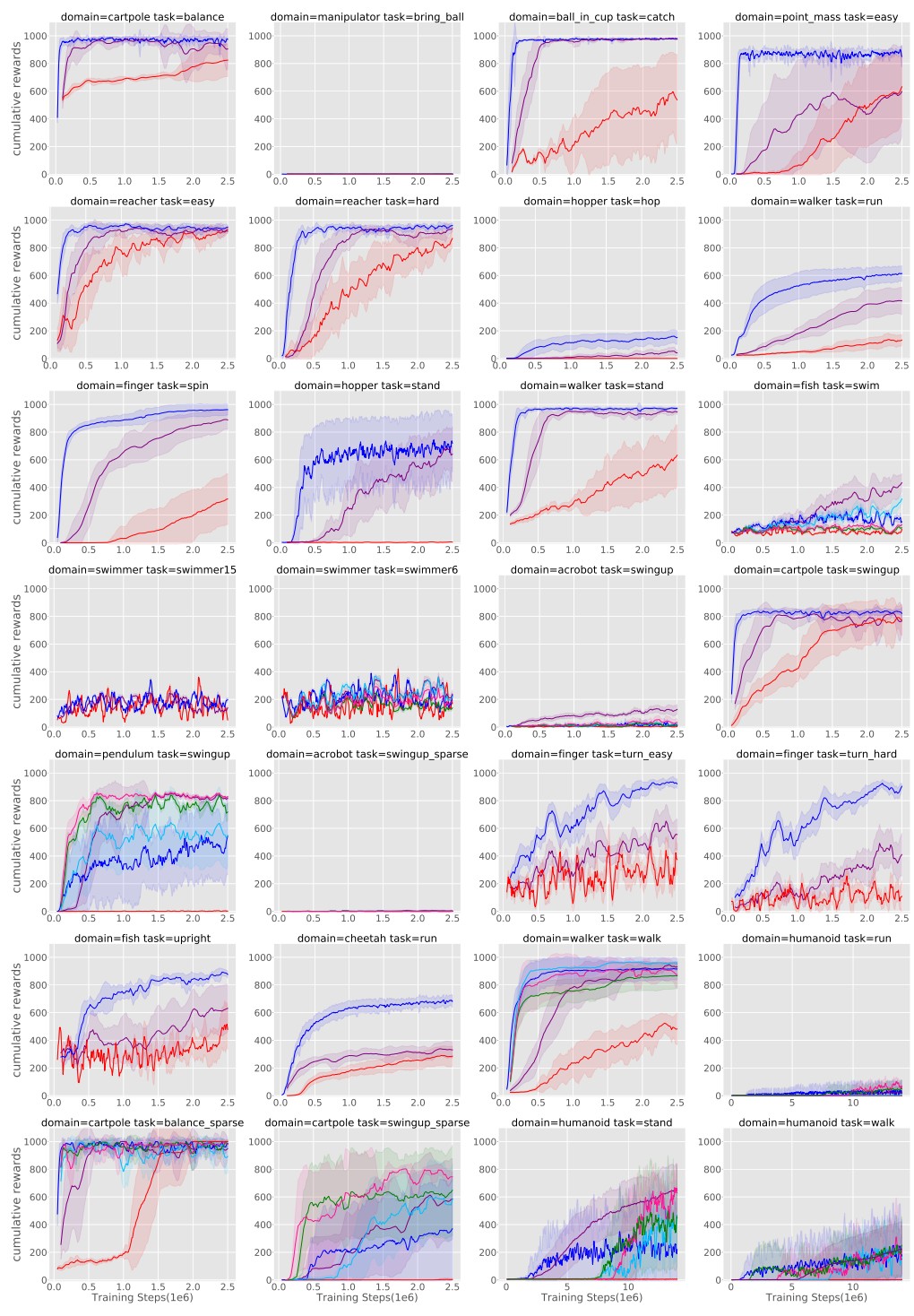

Figure 2: Benchmarking results on DeepMind Control Suite 28 environments. The performances are averaged among 10 random seeds. We use an action repeat of 1 throughout all experiments for simplicity.

In this section, we show the benchmarking results on 28 tasks in DeepMind Control Suite (Figure 2). Each algorithm is run with 2.5M steps (except for humanoid domain; 14M steps), following Abdolmaleki et al. [2]. While the previous work mentioned that tuning the number of action repeats was effective [33], we used an action repeat of 1 throughout all experiments for simplicity. We also use the hyper-parameters of each algorithm

presented in Appendix A. As discussed in Sec. 5.3, we incorporate ELU and layer normalization into SAC in several domains where SAC is behind MPO or AWR. ELU and layer normalization significantly improve performances, especially in pendulum_swingup and cartpole_swingup_sparse. Some of MPO results don't seem to match the original paper, but we appropriately confirmed that these results are equivalent to those of its public implementation [27].

| | SAC | SAC-E$^+$ | SAC-L$^+$ | SAC-E$^+$L$^+$ | MPO | AWR |
|---|---|---|---|---|---|---|
| cartpole_balance | **975 ± 12** | – | – | – | 824 ± 118 | 905 ± 146 |
| manipulator_bring_ball | 0.27 ± 0.0 | 0.80 ± 0.1 | **0.96 ± 0.3** | 0.72 ± 0.1 | 0.79 ± 0.1 | 0.85 ± 0.2 |
| ball_in_cup_catch | **980 ± 0.5** | – | – | – | 976 ± 6 | 538 ± 328 |
| point_mass_easy | **850 ± 75** | – | – | – | 632 ± 254 | 597 ± 329 |
| reacher_easy | **949 ± 14** | – | – | – | 920 ± 18 | 934 ± 25 |
| reacher_hard | **962 ± 18** | – | – | – | 941 ± 22 | 868 ± 70 |
| hopper_hop | **151 ± 50** | – | – | – | 41 ± 22 | 0.1 ± 0.2 |
| walker_run | **615 ± 56** | – | – | – | 416 ± 99 | 133 ± 40 |
| finger_spin | **962 ± 39** | – | – | – | 888 ± 69 | 317 ± 185 |
| hopper_stand | **720 ± 207** | – | – | – | 640 ± 199 | 5 ± 1 |
| walker_stand | **972 ± 6** | – | – | – | 945 ± 18 | 633 ± 221 |
| fish_swim | 152 ± 23 | 317 ± 32 | 108 ± 9 | 130 ± 6 | **434 ± 66** | 97 ± 13 |
| swimmer_swimmer15 | **199 ± 15** | – | – | – | 139 ± 14 | 52 ± 4 |
| swimmer_swimmer6 | 229 ± 12 | 223 ± 11 | 138 ± 13 | 189 ± 17 | **238 ± 29** | 170 ± 4 |
| acrobot_swingup | 10 ± 10 | 21 ± 10 | 15 ± 7 | 34 ± 27 | **127 ± 36** | 4 ± 2 |
| cartpole_swingup | **822 ± 45** | – | – | – | 776 ± 109 | 767 ± 106 |
| pendulum_swingup | 542 ± 279 | 550 ± 232 | 718 ± 62 | **830 ± 4** | 819 ± 11 | 1 ± 4 |
| acrobot_swingup_sparse | 0.40 ± 0.1 | 0.43 ± 0.2 | 0.46 ± 0.0 | 0.42 ± 0.1 | **4 ± 4** | 0.0 ± 0.0 |
| finger_turn_easy | **922 ± 34** | – | – | – | 556 ± 116 | 374 ± 117 |
| finger_turn_hard | **904 ± 21** | – | – | – | 410 ± 156 | 109 ± 113 |
| fish_upright | **876 ± 25** | – | – | – | 631 ± 166 | 478 ± 143 |
| cheetah_run | **682 ± 44** | – | – | – | 331 ± 60 | 285 ± 73 |
| walker_walk | 916 ± 77 | **960 ± 7** | 866 ± 93 | 875 ± 97 | 931 ± 25 | 482 ± 115 |
| humanoid_run | 17 ± 47 | 3 ± 2 | 52 ± 62 | **71 ± 39** | 22 ± 42 | 0.8 ± 0.0 |
| cartpole_balance_sparse | 987 ± 27 | 892 ± 119 | 982 ± 30 | 982 ± 10 | 949 ± 76 | **1000 ± 0.0** |
| cartpole_swingup_sparse | 370 ± 370 | 582 ± 261 | 648 ± 324 | **745 ± 36** | 585 ± 294 | 3 ± 10 |
| humanoid_stand | 221 ± 231 | 469 ± 245 | 448 ± 359 | 630 ± 129 | **651 ± 183** | 6 ± 0.0 |
| humanoid_walk | 182 ± 255 | 166 ± 148 | **250 ± 209** | 219 ± 177 | 224 ± 197 | 1 ± 0.0 |

Table 10: Raw scores of Figure 2. The performances are averaged among 10 random seeds. Each algorithm is run with 2.5M steps (except for humanoid domain; 14M steps), following Abdolmaleki et al. [2]. We use an action repeat of 1 throughout all experiments for simplicity.

# D    Benchmarks on MuJoCo Manipulation Tasks

We extensively evaluate their performance in the manipulation tasks (Figure 3). The trend seems the same as the locomotion tasks, while AWR beats SAC and MPO in Striker, which means they fall into sub-optimal.

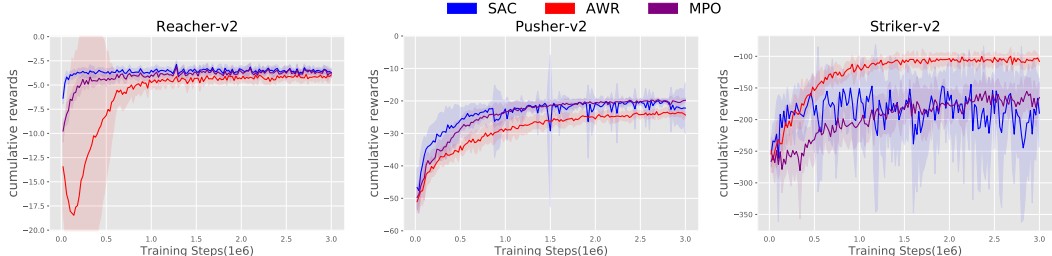

Figure 3: Benchmarking results on OpenAI Gym MuJoCo manipulation environments. All experiments are run with 10 random seeds. SAC and MPO completely solve Reacher and Pusher, while in Striker they fall into sub-optimal.

# E    Reproduction Results of AWR on MuJoCo Locomotion Environments

We re-implemented AWR based on PFRL, a pytorch-based RL library [14], referring its original implementation [48]. Figure 4 shows the performance of our implementation in the original experimental settings, also following hyper-parameters. We recovered the original results in Peng et al. [48] properly.

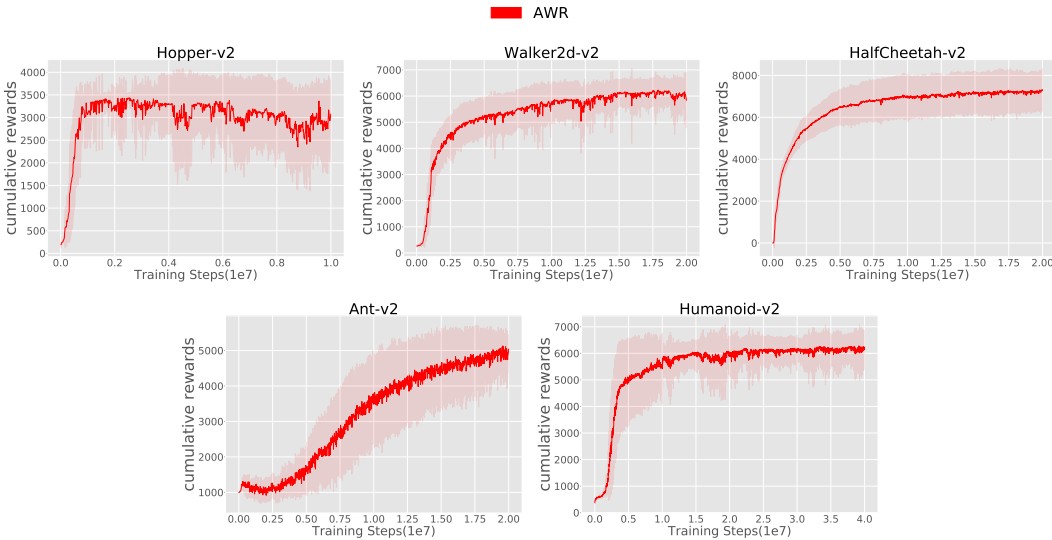

Figure 4: Reproduction of Advantage Weighted Regression (AWR). We obtained comparable results to the original paper.

# F Learning Curves

In this section, we present learning curves of the experiments in Sec. 5.

## F.1 Clipped Double Q-Learning

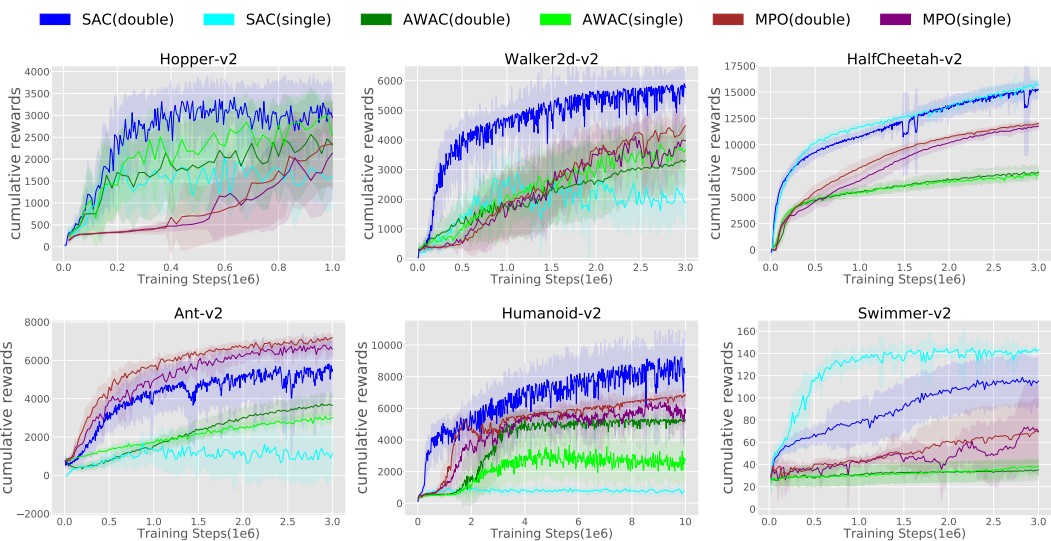

Figure 5: The learning curves of Table 2; ablation of Clipped Double Q-Learning. We test original SAC (double), AWAC (double), MPO (single), and some variants; SAC without clipped double Q-learning (single), AWAC (single), and MPO with clipped double Q-learning (double).

## F.2 Action Distribution for the Policy

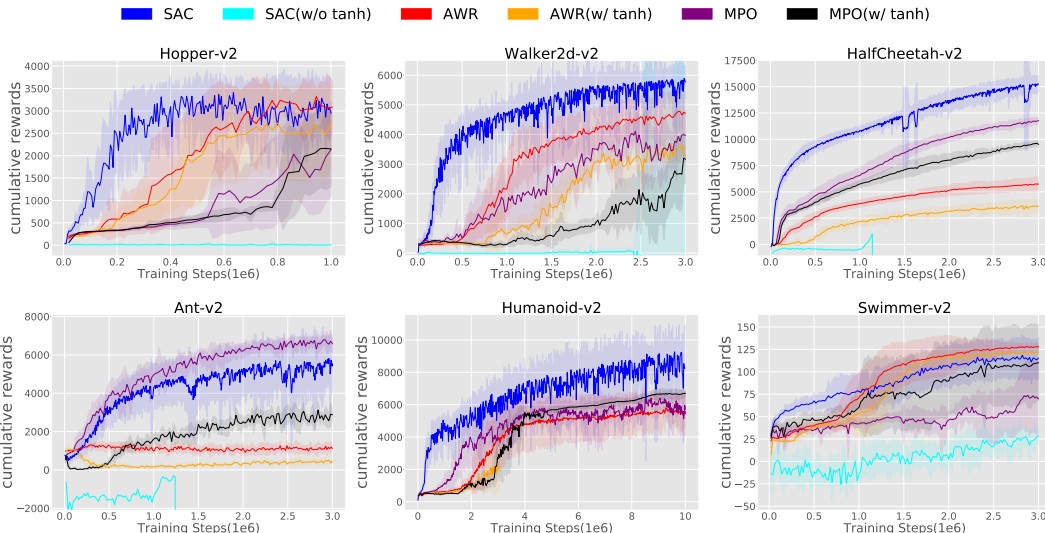

Figure 6: The learning curves of Table 3; ablation of Tanh transformation. A line that stopped in the middle means that its training has stopped at that step due to numerical error. We test SAC without tanh squashing, AWR with tanh, and MPO with tanh.

## F.3 Activation and Normalization

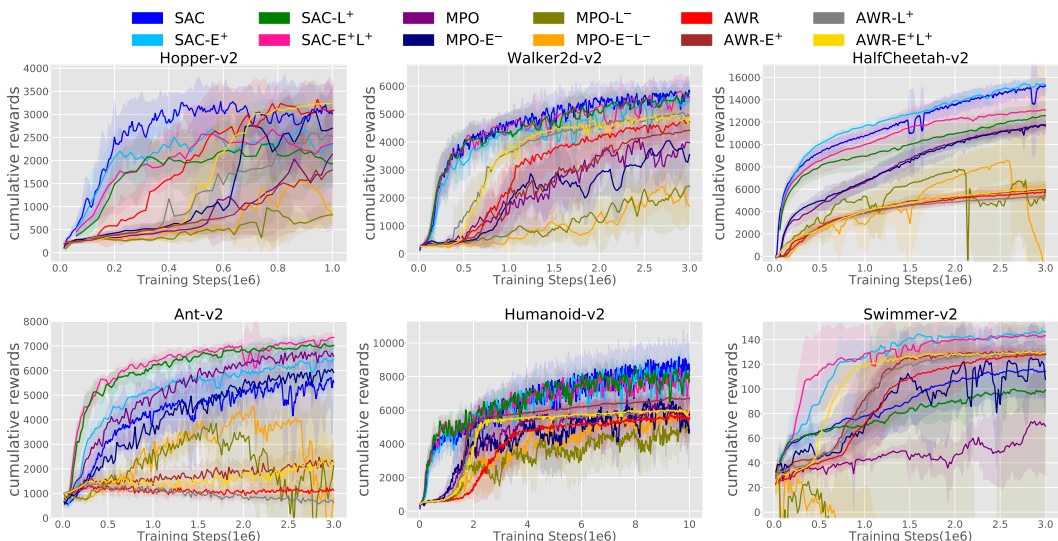

Figure 7: The learning curves of Table 4; incorporating ELU/layer normalization into SAC and AWR. $E^+/L^+$ indicates adding, and $E^-/L^-$ indicates removing ELU/layer normalization.

## F.4 Network Size

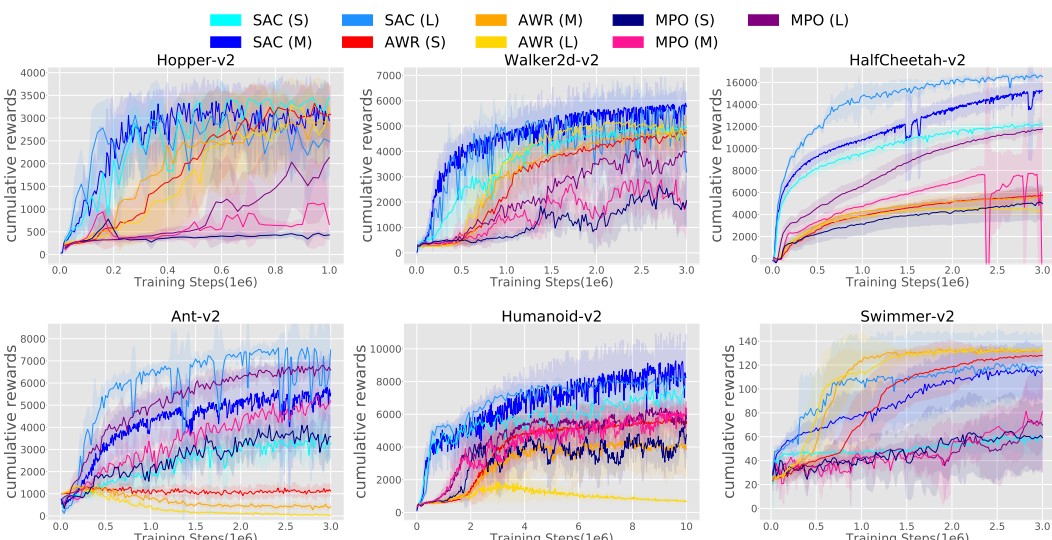

Figure 8: The learning curves of Table 5; experiment for finer network sizes. (S) stands for the small network size from AWR, (M) for the middle network size from SAC, and (L) for the large network size from MPO.

# G    Additional Experiments for Deeper Analysis of Implementation Details

We share the additional experimental results for deeper analysis of implementation details and co-adaptation nature. We report the final cumulative return after 3M steps for Ant/HalfCheetah/Walker2d/Swimmer, 1M steps for Hopper, and 10M steps for Humanoid. All results below are averaged among 10 random seeds. These extensive experimental observations below suggest not only the co-adaptive nature and transferability of each implementation and code detail (discussed in Sec. 6), but also the properties of each kind of algorithm (KL-based and EM-based); **KL-based methods, such as SAC, shows the co-dependent nature in implementation details (clipped double Q-learning, Tanh-Gaussian Policy) but robustness to the code details related to neural networks. In contrast, EM-based methods, such as MPO and AWR, show the co-dependent nature in code details but robustness to the implementation details.** We hope these empirical observations from our experiments are valuable contributions to the RL community.

$\pi_p$ **Update**    We test different types of $\pi_p$ Update as summarized in Table 1 to investigate the effectiveness of implementation choices. We prepare 4 variants: (1) MPO or AWAC with a uniform prior, (2) SAC with target policy instead of a fixed uniform prior, (3) MPO without trust-region (only SG update). The details of (1) - (3) are described below:

(1) We use the actions sampled from uniform distribution as well as the samples from the policy at the past iteration $\pi_{\theta_p^{(k-1)}}$ for the M-step in EM-controls; $a_j \sim \alpha \text{Unif.} + (1 - \alpha)\pi_{\theta_p^{(k-1)}}, \alpha \in (0, 1]$. These variants are much closer to SAC (using uniform distribution as $\pi_p$). We test $\alpha = 0.25, 0.5, 0.75$ for both MPO and AWAC.

(2) We copy the parameter of $\pi_q$ at a certain interval and use it as $\pi_p$ in the objective of KL Control, similar to MPO/PPO/TRPO. It seems "KL-regularized" actor-critic, rather than "soft" (entropy-regularized). We test both Lagrangian constraint ($\epsilon = 0.1, 0.01, 0.001$) and regularization coefficient ($\eta = 1.0, 0.1, 0.01$).

(3) Original MPO stabilizes the $\pi_p$ Update incorporating TR (trust-region) into SG. We test the effect of TR, just removing TR term in the M-step of MPO.

However, the variants listed above have shown drastic degradation compared to the original choice (we omit the performance table since most of them failed). For example, the larger $\alpha$ (1) we chose, the lower scores the algorithm achieved. Also, KL-SAC (2) did not learn meaningful behaviors, and removing TR from MPO (3) induced significant performance drops. These failures suggest that the implementation choice of $\pi_p$ Update might be the most important one and should be designed carefully for both KL and EM control families.

$\mathcal{G}$: **Soft Q-function**    We investigate the effect of the soft Q-function, instead of standard Q function as MPO or AWAC use. We prepare MPO with soft Q, AWAC with soft Q, and SAC without soft Q-function, just modifying Bellman equation and keep the policy objectives as they are.

Table 11 shows that SAC without soft Q degrades its performance over 5 tasks except for Ant, while it is not so drastic compared to clipped double Q or Tanh-Gaussian policy. In contrast, MPO with soft Q slightly improves the performance (over 4 tasks), and AWAC with soft Q slightly also does (over 3 tasks). These trends are similar to the clipped double Q or Tanh-Gaussian policy. We think these experiments support our empirical observation: KL-based methods, such as SAC, show the robustness to the code details, while EM-based methods, such as MPO and AWR, show the co-dependent nature in code details but robustness to the implementation details.

|                   | Hopper-v2        | Walker2d-v2      | HalfCheetah-v2     | Ant-v2           | Humanoid-v2       | Swimmer-v2     |
|-------------------|------------------|------------------|--------------------|------------------|-------------------|----------------|
| **SAC**           | $3013 \pm 602$   | $5820 \pm 411$   | $15254 \pm 751$    | $5532 \pm 1266$  | $8081 \pm 1149$   | $114 \pm 21$   |
| **SAC (w/o Soft Q)** | $2487 \pm 870$ | $5674 \pm 202$   | $12319 \pm 2731$   | $6496 \pm 305$   | $6772 \pm 3060$   | $114 \pm 33$   |
| **MPO**           | $2136 \pm 1047$  | $3972 \pm 849$   | $11769 \pm 321$    | $6584 \pm 455$   | $5709 \pm 1081$   | $70 \pm 40$    |
| **MPO (w/ Soft Q)** | $2271 \pm 1267$ | $3817 \pm 794$  | $11911 \pm 274$    | $6312 \pm 332$   | $6571 \pm 461$    | $80 \pm 32$    |
| **AWAC**          | $2329 \pm 1020$  | $3307 \pm 780$   | $7396 \pm 677$     | $3659 \pm 523$   | $5243 \pm 200$    | $35 \pm 8$     |
| **AWAC (w/ Soft Q)** | $2545 \pm 1062$ | $3671 \pm 575$ | $7199 \pm 628$     | $3862 \pm 483$   | $5152 \pm 162$    | $35 \pm 10$    |

Table 11: Ablation of Soft Q-function (the choice of $\mathcal{G}$ in Table 1), adding to MPO and AWAC while removing from SAC.

**Network Size for AWAC**    To investigate the co-dependent nature between implementation and code details more precisely, we add the network size ablation of AWAC, whose implementations stand between MPO and AWR. AWAC differs $\pi_p$ Update and network size (the default choice of AWAC is (M)) from MPO (in fact, MPO uses TD(0) in open-source implementation [27] and we assume the difference of $\pi_\theta$ might be minor). Also, AWAC differs $\mathcal{G}$ and $\mathcal{G}$ estimate from AWR.

The results of AWAC (Table 12) show a similar trend to AWR in high-dimensional tasks (Ant, Humanoid); a larger network did not help. We may hypothesize that $\pi_p$ Update of AWR/AWAC, mixture+SG, is not good at optimizing larger networks, compared to SG + TR of MPO. In contrast, especially, Hopper and Walker2d show a

similar trend to MPO; the larger, the better. Totally, AWAC with different network sizes shows the mixture trend of AWR and MPO, which is the same as implementation details. We think these observations might highlight the co-adaptation nature between implementation and code details.

|  | Hopper-v2 | Walker2d-v2 | HalfCheetah-v2 | Ant-v2 | Humanoid-v2 | Swimmer-v2 |
|---|---|---|---|---|---|---|
| **AWAC (L)** | $\mathbf{2764 \pm 919}$ | $\mathbf{4350 \pm 542}$ | $6433 \pm 832$ | $2342 \pm 269$ | $4164 \pm 1707$ | $\mathbf{40 \pm 5}$ |
| **AWAC (M)** | $2329 \pm 1020$ | $3307 \pm 780$ | $\mathbf{7396 \pm 677}$ | $3659 \pm 523$ | $5243 \pm 200$ | $35 \pm 8$ |
| **AWAC (S)** | $2038 \pm 1152$ | $2022 \pm 971$ | $5864 \pm 768$ | $\mathbf{3705 \pm 659}$ | $\mathbf{5331 \pm 125}$ | $34 \pm 11$ |

Table 12: Ablation of network size for AWAC.

**Combination of Clipped Double Q-Learning/Tanh-Gaussian and Soft Q-function**  We observe that both clipped double Q-learning/Tanh-Gaussian policy and soft Q-function are the important implementation choices to KL control, SAC, which lead to significant performance gains. To test the co-adaptation nature more in detail, we implement these two choices into MPO and AWAC at the same time.

The results (Table 13 and Table 14) show that incorporating such multiple combinations does not show any notable improvement in EM Controls, MPO and AWAC. They also suggest the co-adaptation nature of those two implementations to KL Controls, especially SAC.

|  | Hopper-v2 | Walker2d-v2 | HalfCheetah-v2 | Ant-v2 | Humanoid-v2 | Swimmer-v2 |
|---|---|---|---|---|---|---|
| **MPO (S)** | $2136 \pm 1047$ | $3972 \pm 849$ | $11769 \pm 321$ | $6584 \pm 455$ | $5709 \pm 1081$ | $70 \pm 40$ |
| **MPO (D)** | $2352 \pm 959$ | $\mathbf{4471 \pm 281}$ | $12028 \pm 191$ | $\mathbf{7179 \pm 190}$ | $6858 \pm 373$ | $69 \pm 29$ |
| **MPO (Soft Q, S)** | $2271 \pm 1267$ | $3817 \pm 794$ | $11911 \pm 274$ | $6312 \pm 332$ | $6571 \pm 461$ | $\mathbf{80 \pm 32}$ |
| **MPO (Soft Q, D)** | $1283 \pm 632$ | $4378 \pm 252$ | $\mathbf{12117 \pm 126}$ | $6822 \pm 94$ | $\mathbf{6895 \pm 433}$ | $45 \pm 4$ |
| **AWAC (S)** | $2540 \pm 755$ | $3662 \pm 712$ | $7226 \pm 449$ | $3008 \pm 375$ | $2738 \pm 982$ | $38 \pm 7$ |
| **AWAC (D)** | $2329 \pm 1020$ | $3307 \pm 780$ | $7396 \pm 677$ | $3659 \pm 523$ | $5243 \pm 200$ | $35 \pm 8$ |
| **AWAC (Soft Q, S)** | $\mathbf{2732 \pm 660}$ | $3658 \pm 416$ | $7270 \pm 185$ | $3494 \pm 330$ | $2926 \pm 1134$ | $36 \pm 10$ |
| **AWAC (Soft Q, D)** | $2545 \pm 1062$ | $3671 \pm 575$ | $7199 \pm 628$ | $3862 \pm 483$ | $5152 \pm 162$ | $35 \pm 10$ |

Table 13: Ablation of combination in implementation components; Soft Q-function (the choice of $\mathcal{G}$) and Clipped Double Q-Learning (the choice of $\mathcal{G}$ estimate), adding to MPO and AWAC. (D) denotes algorithms with clipped double Q-learning, and (S) denotes without it.

|  | Hopper-v2 | Walker2d-v2 | HalfCheetah-v2 | Ant-v2 | Humanoid-v2 | Swimmer-v2 |
|---|---|---|---|---|---|---|
| **MPO** | $2136 \pm 1047$ | $\mathbf{3972 \pm 849}$ | $11769 \pm 321$ | $\mathbf{6584 \pm 455}$ | $5709 \pm 1081$ | $70 \pm 40$ |
| **MPO (Soft Q)** | $2271 \pm 1267$ | $3817 \pm 794$ | $\mathbf{11911 \pm 274}$ | $6312 \pm 332$ | $\mathbf{6571 \pm 461}$ | $\mathbf{80 \pm 32}$ |
| **MPO (Soft Q, Tanh)** | $314 \pm 8^{\dagger}$ | $368 \pm 47^{\dagger}$ | $3427 \pm 207^{\dagger}$ | $628 \pm 221^{\dagger}$ | $5919 \pm 202^{\dagger}$ | $35 \pm 8^{\dagger}$ |
| **AWAC** | $2329 \pm 1020$ | $3307 \pm 780$ | $7396 \pm 677$ | $3659 \pm 523$ | $5243 \pm 200$ | $35 \pm 8$ |
| **AWAC (Soft Q)** | $2545 \pm 1062$ | $3671 \pm 575$ | $7199 \pm 628$ | $3862 \pm 483$ | $5152 \pm 162$ | $35 \pm 10$ |
| **AWAC (Soft Q, Tanh)** | $\mathbf{2989 \pm 484}$ | $2794 \pm 1692$ | $6263 \pm 247$ | $3507 \pm 458$ | $66 \pm 4$ | $32 \pm 5$ |

Table 14: Ablation of combination in implementation components; Soft Q-function (the choice of $\mathcal{G}$) and Tanh-squashed Gaussian policy (the parameterization of the policy), adding to MPO and AWAC ($^{\dagger}$numerical error happens during training).

# H Failed Ablations

This section provides the failure case of ablations on tanh-squashed distributions and exchanging network architectures, which shows the catastrophic failure during training, and unclear insights.

## H.1 Action Distribution for the Policy: Without Action Clipping

We observe that naive application of tanh-squashing to MPO and AWR significantly suffers from numerical instability, which ends up with NaN outputs (Table 15 and Figure 9). As we point out in Sec. 5.2, the practical solution is to clip the action within the supports of distribution surely; $a \in [-1 + \epsilon, 1 - \epsilon]^{|\mathcal{A}|}$.

```
eps = 1e-6
actions = torch.clamp(actions, min=-1.+eps, max=1.-eps)
```

| | SAC (w/) | SAC (w/o) | AWR (w/) | AWR (w/o) | MPO (w/) | MPO (w/o) |
|---|---|---|---|---|---|---|
| Hopper-v2 | $3013 \pm 602$ | $6 \pm 10$ | $\mathbf{3267 \pm 383}$ | $3085 \pm 593$ | $301 \pm 12^{\dagger}$ | $2136 \pm 1047$ |
| Walker2d-v2 | $\mathbf{5820 \pm 411}$ | $-\infty$ | $3281 \pm 1084^{\dagger}$ | $4717 \pm 678$ | $328 \pm 95^{\dagger}$ | $3972 \pm 849$ |
| HalfCheetah-v2 | $\mathbf{15254 \pm 751}$ | $-\infty$ | $1159 \pm 599^{\dagger}$ | $5742 \pm 667$ | $831 \pm 242^{\dagger}$ | $11769 \pm 321$ |
| Ant-v2 | $5532 \pm 1266$ | $-\infty$ | $152 \pm 101^{\dagger}$ | $1127 \pm 224$ | $202 \pm 102^{\dagger}$ | $\mathbf{6584 \pm 455}$ |
| Humanoid-v2 | $\mathbf{8081 \pm 1149}$ | $108 \pm 82^{\dagger}$ | $538 \pm 49^{\dagger}$ | $5573 \pm 1020$ | $5642 \pm 77^{\dagger}$ | $5709 \pm 1081$ |
| Swimmer-v2 | $114 \pm 21$ | $28 \pm 11$ | $117 \pm 16$ | $\mathbf{128 \pm 4}$ | $37 \pm 6^{\dagger}$ | $70 \pm 40$ |

Table 15: Ablation of Tanh transformation ($^{\dagger}$numerical error happens during training). We test SAC without tanh squashing, AWR with tanh, and MPO with tanh. SAC without tanh transform results in drastic degradation of the performance, which can be caused by the maximum entropy objective that encourages the maximization of the covariance.

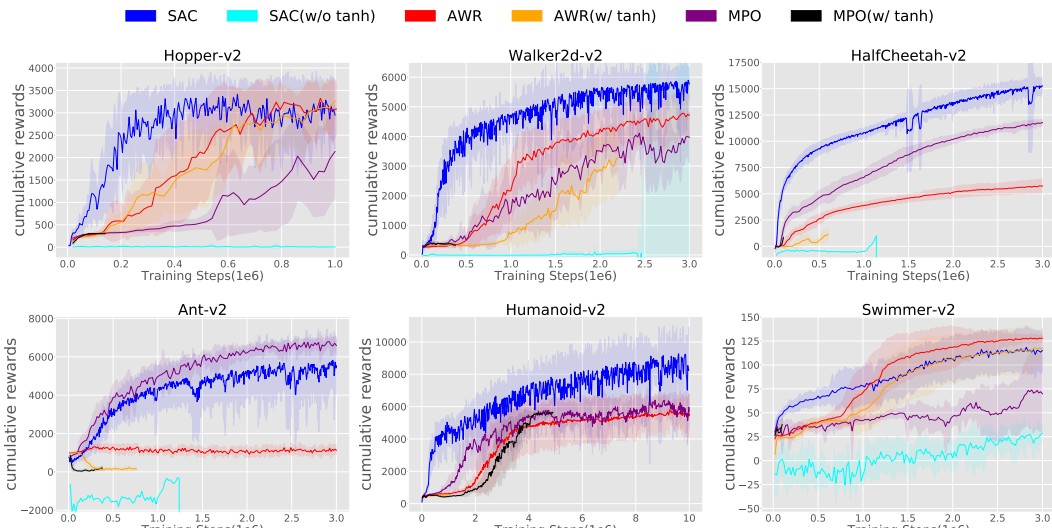

Figure 9: The learning curves of Table 15. We test SAC without tanh-squashed distribution, AWR with tanh, and MPO with tanh. SAC without tanh transform (using MPO action penalty instead) results in drastic degradation of the performance, which can be caused by the maximum entropy objective that encourages the maximization of the covariance. AWR and MPO with tanh squashing become numerically unstable. A line that stopped in the middle means that its training has stopped at that step due to numerical error.

## H.2 Network Architecture: Whole Swapping

In contrast to prior works on TRPO and PPO, the network architecture that works well in all the off-policy inference-based methods is not obvious, and the RL community doesn't have an agreeable default choice. Since the solution space is too broad without any prior knowledge, one possible ablation is that we test 3 different architectures that work well on at least one algorithm.

To validate the dependency of the performance on the network architecture, we exchange the configuration of the policy and value networks, namely, the size and number of hidden layers, the type of activation function, network optimizer and learning rate, weight-initialization, and the normalization of input state (See Appendix A). All other components remain the original implementations.

However, this ablation study might end up the insufficient coverage and the unclear insights. We broke down the network architecture comparison into the one-by-one ablations of activation and normalization, and experimented with finer network sizes.

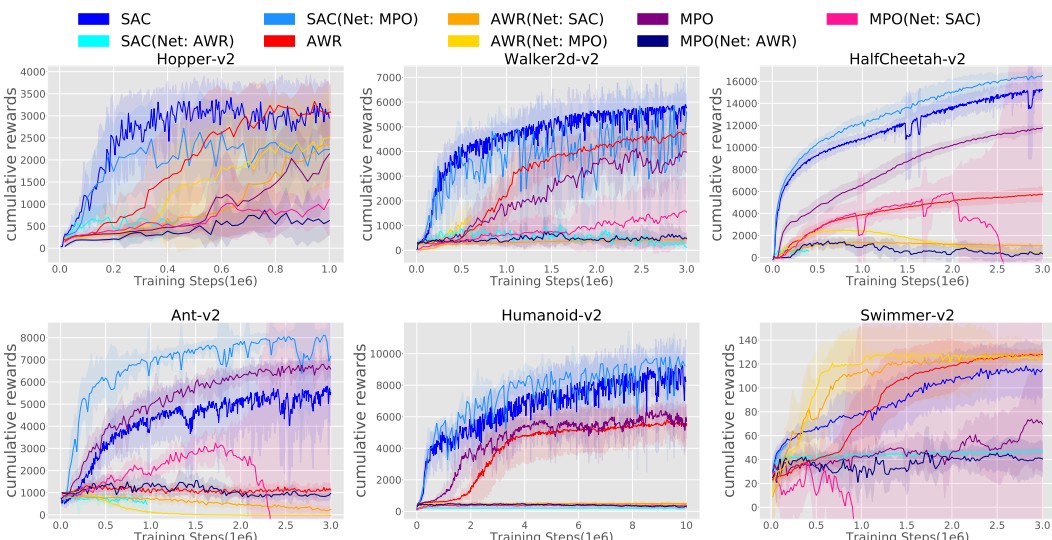

Figure 10: Swapping network architectures between each methods. These results suggest that these off-policy inference-based algorithms might be fragile with other network architectures and more co-dependent with architectures than on-policy algorithms. A line that stops in the middle means that training has stopped at that step with numerical error due to NaN outputs.

| Algorithm | Architecture | | |
|---|---|---|---|
| | **MPO** | **AWR** | **SAC** |
| MPO | $2136 \pm 1047$ | $623 \pm 316$ | $1108 \pm 828$ |
| AWR | $2509 \pm 1117$ | $\mathbf{3085 \pm 593}$ | $2352 \pm 960$ |
| SAC | $2239 \pm 669$ | $651 \pm 381^{\dagger}$ | $3013 \pm 602$ |

Table 16: Results in Hopper-v2 environment ($^{\dagger}$numerical error happens during training). All results are averaged over 10 seeds and we also show their standard deviations.

| Algorithm | Architecture | | |
|---|---|---|---|
| | **MPO** | **AWR** | **SAC** |
| MPO | $3972 \pm 849$ | $481 \pm 210$ | $1548 \pm 1390$ |
| AWR | $1312 \pm 680^{\dagger}$ | $4717 \pm 678$ | $428 \pm 89$ |
| SAC | $5598 \pm 795$ | $117 \pm 164$ | $\mathbf{5820 \pm 556}$ |

Table 17: Results in Walker2d-v2 environment ($^{\dagger}$numerical error happens during training). All results are averaged over 10 seeds and we also show their standard deviations.

| Algorithm | Architecture | | |
|---|---|---|---|
| | **MPO** | **AWR** | **SAC** |
| MPO | $11769 \pm 321$ | $339 \pm 517$ | $-\infty$ |
| AWR | $485 \pm 57$ | $5742 \pm 667$ | $1060 \pm 146$ |
| SAC | $\mathbf{16541 \pm 341}$ | $589 \pm 367^{\dagger}$ | $15254 \pm 751$ |

Table 18: Results in HalfCheetah-v2 environment ($^{\dagger}$numerical error happens during training). All results are averaged over 10 seeds and we also show their standard deviations.

| Algorithm | Architecture | | |
|---|---|---|---|
| | **MPO** | **AWR** | **SAC** |
| MPO | $6584 \pm 455$ | $967 \pm 202$ | $-\infty$ |
| AWR | $-30 \pm 12$ | $1127 \pm 224$ | $243 \pm 167$ |
| SAC | $\mathbf{7159 \pm 1577}$ | $479 \pm 463^{\dagger}$ | $5532 \pm 1266$ |

Table 19: Results in Ant-v2 environment ($^{\dagger}$numerical error happens during training). All results are averaged over 10 seeds and we also show their standard deviations.

| Algorithm | Architecture | | |
|---|---|---|---|
| | **MPO** | **AWR** | **SAC** |
| MPO | $5709 \pm 1081$ | $288 \pm 126$ | $371 \pm 72$ |
| AWR | $420 \pm 30$ | $5573 \pm 1020$ | $507 \pm 48$ |
| SAC | $\mathbf{9225 \pm 1010}$ | $205 \pm 0$ | $8081 \pm 1149$ |

Table 20: Results in Humanoid-v2 environment. All results are averaged over 10 seeds and we also show their standard deviations.

| Algorithm | Architecture | | |
|---|---|---|---|
| | **MPO** | **AWR** | **SAC** |
| MPO | $70 \pm 40$ | $41 \pm 15$ | $-\infty$ |
| AWR | $124 \pm 3$ | $128 \pm 4$ | $\mathbf{130 \pm 8}$ |
| SAC | $53 \pm 6^{\dagger}$ | $47 \pm 3$ | $114 \pm 21$ |

Table 21: Results in Swimmer-v2 environment ($^{\dagger}$numerical error happens during training). All results are averaged over 10 seeds and we also show their standard deviations.