# OpenReview forum: "Co-Adaptation of Algorithmic and Implementational Innovations in Inference-based Deep Reinforcement Learning"
_NeurIPS.cc/2021/Conference — NeurIPS 2021 Poster_

### Official Review · Reviewer_34Q6 · 2021-07-13

**Rating:** 6
**Confidence:** 3

**Summary:**

Comparing the performance of different RL algorithms can sometimes be difficult because algorithmic contributions can be confounded with differing implementational choices between the agents. This paper attempts to examine the relative contributions of algorithmic vs implementational choices in a series of inference-based, off-policy actor-critic algorithms -- MPO, AWR, and SAC. The authors' starting point is a unified view of these algorithms from the perspective of control-as-inference. In this framework, policies as cast as posterior probabilities that a given action in a given state will be optimal. The authors derive an ELBO for this posterior that, when optimised, yields an approximate optimal policy. The authors then describe how optimization can be approached in two ways: an EM approach (where both a prior and posterior policy are optimized) and a KL approach (where only the posterior policy is optimized). MPO and AWR belong to the EM family of algorithms, while SAC belongs to the KL family. The authors further detail how these algorithms fit into their unified framework.

In the second half of the paper, the authors focus on the implementational differences between these algorithms. They focus specifically on four choices: clipped double q-learning, tanh-squashing in the policies, the choice of activation function and layer norm, and the network size. Specifically, through a series of ablation studies, the authors examine how critical some of these choices are to specific algorithms and which of these can be transferred to other algorithms.

**Limitations And Societal Impact:**

Given that the foundational nature of this work, there would not be any malignant implications beyond that which is already inherent in basic RL, so a discussion of the potential societal impact is not relevant for this work.

**Main Review:**

Overall, this was an interesting paper. For me, one of the paper's strengths was its theoretical framework casting MPO, AWR, SAC and other algorithms as special cases of a more unified control-as-inference objective, and the discussion of EM control vs KL control was rather enlightening. It was unclear to me, however, whether the authors were regarding the "Implementational Details" of Section 4.2 as possible transferable implementational choices, on the same level as activation functions and network sizes, or whether these were to be interpreted as different algorithms.

My main hesitations, however, lie with the empirical side of this paper, and for the reasons below, I do not believe the paper yet ready for acceptance:
* the authors ablated the different implementational choices individual, but my concern is that there may be co-dependencies between different implementational choices, and individual ablations may fail to capture that. For instance, whether the tanh-squashing helps or hurts the policy network depends on whether and how the policies are regularised, something which may be more pertinent to KL control rather than EM.
* there also seems to be a thematic disconnect between the four implementational choices the authors have focused on and the preceding control-as-inference discussion. The four implementational choices do not seem to depend in any way on the fact that MPO, AWR, and SAC can be treated in a unified, inference-based manner. Whether to use layer norm, ELU or RELU, large or small networks, tanh squashing, or double Q-learning is a question that applies equally well to any deep RL algorithm, not just the ones discussed here. Hence it is puzzling why the authors chose these four differences to focus on given the topic of their paper. Furthermore, if the differences in Section 4.2 can be regarded as implementational differences, it would have been more interesting and thematically-consistent with the first half of this paper to examine the transferability of those choices instead.
* for algorithms that originally included a certain implementational choice, it is clear that ablation leads to a decline in performance. However, I find in several cases that the authors' claim of improved performance less convincing. For instance, the authors claim that using ELU and layer norm across all algorithms will improve performance. While doing this certainly didn't hurt performance and even led to a higher mean score in some cases, the score increases do not seem to reach the level of statistical significance given the size of the standard deviations.
* finally, the scope of the empirical studies seemed a bit limited to me. The authors only looked at ablating four implementational choices, and it would have strengthened the paper significantly had the authors examined a wider selection (see e.g. Hessel et al, 2017 for an example of a more extensive ablation study in RL).

Minor errors:
* the authors reference equation 3 several times (e.g. lines 187 and 203), but this equation is never numbered in the main text. Presumably, it is the equation that appears in the M-step, after line 157.
* typo in line 268:  “achieve” rather than “archive”.
* typo in 294: “agreed upon default choice” rather than “agreeable default choice”.





--- Review update ---

After further consideration and based on the clarification and additional results the authors have provided, I have decided to upgrade my score to 6. To me, this work offers two very valuable contributions to the community. The first is the theoretical demonstration that many disparate policy gradient algorithms can actually be seen within a unified control-as-inference framework, a framing that I find novel and exciting. This unified framework not only provides a deeper understanding of pre-existing algorithm, but it also offers a new means to study and inform the design of future algorithms. The second is, as both reviewer CYmr and the AC have pointed out, the paper's attempt to survey the various algorithms already in the literature using this framework, examining how transferable certain implementational or code components are within the KL or EM families. What the authors are attempting with this work is definitely of great interest to the community.

Given the huge scope, I appreciate that exhaustive exploration on the empirical side is not possible. The new results were rather interesting, but for me, certain questions remain. For instance, the latest results have not entirely addressed my question about co-dependencies. From their limited transfer studies, the authors have tried to draw general conclusions about transferability between the KL or EM families; yet, I still wonder whether there are synergies between components that the authors break with their ablations and whether some of the performance drop has more to do with breaking these synergies than whether the algorithm is KL or EM. Again, as an example, I mention the possibility that soft-Q functions may synergise more with tanh-gaussian policies and clipped double q-learning. In the ablation studies, the authors found that using tanh-gaussianity and/or clipped double q-learning didn't transfer well to EM algorithms. But this could be because tanh-gaussianity and double clipped q-learning work best with a soft-Q function, and these components performed worse because soft-Q  was absent. This is just one example of how possible co-dependencies may confound the conclusions that the authors sought to make in their ablation studies. Regarding the new results, the failure case of alternative $\pi_p$ was also rather intriguing. In short, there is a lot in the space of algorithms that remains to be explored, but doing an exhaustive exploration would be an enormous undertaking.

In the end, I am inclined to support accepting support of the paper not only for the reasons I mentioned at the beginning, but also because I believe doing so will provoke much important discussion and inspire new research to address the questions that may come out of it.

**Time Spent Reviewing:**

6

---

> ### Author Response · Authors · 2021-08-10
> **Author Response**
>
> We appreciate the reviewer for the careful reading of the paper and constructive comments for improving the quality.
>
> **> Clarification of implementation details:**
> As the reviewer pointed out, we will explicitly separate current **“implementation details”** into **“implementation and code details”** in the revision, where new “implementation details” includes the components listed up in Table 1 ($\mathcal{G}$ estimate: clipped double Q-learning, $\pi_{\theta}$: Tanh-Gaussian Policy, in the experiments), and “code details” includes the details of neural network (network size, activation function, normalization), that is also a very important aspect of modern deep RL research as the previous empirical paper suggests (Andrychowicz et al. (2021), Engstrom et al. (2020), Henderson et al. (2018), Islam et al. (2017)).
>
> Our extensive experimental observation suggests not only the co-adaptive nature and transferability of each implementation and code detail (discussed in Section 6, Table 6), but also the properties of each kind of algorithm (KL-based and EM-based); **KL-based methods, such as SAC, shows the co-dependent nature in implementation details (clipped double Q-learning, Tanh-Gaussian Policy) but robustness to the code details. In contrast, EM-based methods, such as MPO and AWR, show the co-dependent nature in code details but robustness to the implementation details.** We believe these empirical observations from our experiments are valuable contributions to the RL community.
>
>
> **> The choice of ablation study:**
> Our empirical study widely covers both implementational (clipped double Q-learning, Tanh-Gaussian Policy) and code (network size, activation function, normalization) choices to investigate what aspect of each algorithm has an effect on the performance in the benchmark environments. As described in Section 2, while most of the prior meta-analysis papers have only focused on the single family of on-policy algorithms (e.g. TRPO and PPO), we treat two distinct families of off-policy algorithms from a unified perspective and compare them mathematically and empirically, to investigate each algorithmic nature.
>
> To remove the co-dependencies in the code, we first tried to normalize the architecture of each algorithm, simply exchanging the configuration of the policy and value networks, namely, the size and number of hidden layers, the type of activation function, network optimizer and learning rate, weight-initialization, and the normalization of input state, while keeping the original implementations in the other components (See Appendix G; Failed Ablations).
>
> However, while it is a common approach in on-policy comparison, this forced normalization might end up the insufficient coverage and unclear insights due to the off-policy instability; it is better to keep original implementations. We broke down the network architecture comparison into the one-by-one ablations of activation and normalization, and experimented with finer network sizes.
> These experimental designs enable us to successfully find an important insight: ELU and layer normalization are transferable to SAC and AWR, and KL-based methods are robust to code details, while EM-based methods are robust to implementation details. We believe our ablation choices effectively highlight the algorithmic properties.
>
>
>
> **> ELU and Layer Normalization:**
> In addition to Section 5.3, we also tested the ablation on the activation function and normalization in the neural network, with 12 DM Control tasks where the performances of SAC is behind those of MPO (See Figure 2 and Table 10 in Appendix C), and shows the applicability to the SAC improving the performance especially in pendulum_swingup, cartpole_swingup_sparse, and humanoid_stand. We believe that it is worth considering replacing the activation function from ReLU to ELU and incorporating the layer normalization and that it seems a more transferable choice rather than implementation details (clipped double Q-learning and Tanh-Gaussian Policy) in EM-based methods (MPO and AWR) .
>
>
> **> Experimental Coverage:**
> We thank the reviewer for providing the additional reference to the experiment. Related to this work, we’d like to clarify our motivation for the experimental evaluation. While Hessel et al. (2017) examined the better combination of DQN extensions in the previous works, and proposed the best algorithm, Rainbow, our work focuses on not proposing best-performed algorithms, but revealing the effect of each implementation and code and algorithmic properties that each family of the algorithm has, which has been often overlooked and less discussed in the algorithmic papers. Our unified formulation in Section 3 & 4 allows us such implementation and code comparison, and we successfully reveal the nature of both KL-based and EM-based algorithms, discussed in Section 5 & 6. As a byproduct of these comparisons, we may serve several practical recommendations for the practitioners.
>
> In addition, we’d like to know more concretely what kind of comparison isn’t sufficient for now. Could the reviewer let us know the ablation or evaluation that you think are important or should be included (e.g. another implementation details, or finer grid search of the current experiments such as network size, etc)?  We have an extensive codebase, and would be happy to address any specific additional ablations that you will suggest for the final version. We might deal with some of them and update the results by the end of the discussion period, the start of September.
>
>
> **> Response to Minor errors:**
> We thank you for pointing out the minor errors in our manuscript. We will fix them appropriately in the revision.
>
> ---
> **Reference**
>
> Andrychowicz et al. (2021). What matters for on-policy deep actor-critic methods? a large-scale study. In International Conference on Learning Representations.
>
> Engstrom et al. (2020). Implementation matters in deep rl: A case study on ppo and trpo. In International conference on learning representations.
>
> Hessel et al. (2018). Rainbow: Combining improvements in deep reinforcement learning. In Thirty-second AAAI conference on artificial intelligence.
>
> Henderson et al. (2018). Deep reinforcement learning that matters. In Thirty-second AAAI conference on artificial intelligence.
>
> Islam et al. (2017). Reproducibility of benchmarked deep reinforcement learning tasks for continuous control. arXiv preprint arXiv:1708.04133.

---

> > ### Comment · Reviewer_34Q6 · 2021-08-18
> > **Clarification of the taxonomy**
> >
> > I thank the authors for their careful response to my comments. After reading their comments and looking at the paper again, I think it might help to start by clarifying their taxonomy of the policy gradient algorithms. I found their treatment of these various algorithms within a unified control-as-inference framework very insightful, but this unification had the unintended effect of blurring in this reviewer's mind what now constitutes an "algorithm" and what constitutes "implementational" or "code" details. There is a clear division between EM and KL-based algorithms, but within each family, what the authors regard as individual algorithms is now unclear. For instance, should everything in the first three rows of Table 1 (MPO, AWC, AWAC) be regarded as implementational details of a generic EM algorithm, where the RL practitioner is free to mix-and-match between the different components to create any novel EM algorithm that they please? Presumably this is not what the authors have mind, as they talk about distinct MPO, AWC, and AWAC algorithms. In which case, what are the parts that define e.g. MPO in distinction to AWC and AWAC, and why should this be regarded as a meaningful definition of MPO? Put another way, why don't the authors just talk about a generic EM algorithm with MPO, AWC, and AWAC now regarded as specific implementations of that? The same question applies for the KL-family of algorithms, although the authors here only looked at a single instance of that -- SAC.

---

> > > ### Author Response · Authors · 2021-08-19
> > > **Clarification and Modification of Table 1**
> > >
> > > We appreciate your feedback on the ambiguity of the taxonomy (Table 1).
> > >
> > > We modified the two sections in the table -- “Algorithm”, “Implementation” -- as follows. We hope our modification clarifies the ambiguity of the taxonomy and deals with your concerns. If not, let us know what you think! It’s been helpful feedback for improving the paper.
> > >
> > > - **Algorithm:** We classify the RL methods based on how to maximize the ELBO “mathematically” (Eq. 1). As you suggested and also as implied by our original Table 1, we only need 2 generic algorithm families (EM or KL controls; general cases are described in Section 4.1.1 and 4.1.2) to categorize 13 popular policy gradient algorithms in Table 1. The differences among RL methods in each group are all in “Implementation” and “Code”, based on our categorization. “RL practitioner is free to mix-and-match between the different components to create any novel EM algorithm that they please” is exactly the concern we want to raise, since many algorithms with seemingly different derivations/stories/popularities are in fact very close mathematically, if we strip away “almost arbitrarily chosen” implementation (or code) details.
> > >
> > > - **Implementation:** Any method that uses trust-region policy update (TRPO/PPO/MPO) is listed with “TR” in “Implementation: $\pi_q$ updates” (see Section 4.2.1 and Appendix B for how “Analytic+TR”, “TR”, and “SG+TR” differ).
> > >
> > >
> > > ---
> > > ### **[Modified Table 1]**
> > > (edited 8/21)
> > >
> > > | *Method* | *Algorithm* | *Imprementation*: $\pi_q$ update | $\pi_p$ update | $\mathcal{G}$ | $\mathcal{G}$ estimate | $\pi_{\theta}$ |
> > > | -------- | -------: | -------: | -------: | -------: | -------: | -------: |
> > > | **MPO** | *EM* | Analytic + TR | SG + TR | $Q^{\pi_p}$ | Retrace(1) | $\pi_p =\mathcal{N}(\mu_{\theta}(s), \Sigma_{\theta}(s))$ |
> > > | **AWR** | *EM* | Analytic | Mixture + SG | $A^{\tilde{\pi}_p}$ | TD($\lambda$) | $\pi_p =\mathcal{N}(\mu_{\theta}(s), \Sigma)$ |
> > > | **AWAC** | *EM* | Analytic | Mixture + SG | $Q^{\pi_p}$   | TD(0) | $\pi_p =\mathcal{N}(\mu_{\theta}(s), \Sigma_{\theta})$ |
> > > | **SAC** | *KL* | SG | (Fixed to Unif.) | $Q^{\pi_q}_{\text{soft}}$ | TD(0) + TD3 | $\pi_q = \text{Tanh}$($\mathcal{N}(\mu_{\theta}(s), \Sigma_{\theta}(s))$) |
> > > | **PoWER** | *EM* | Analytic | Analytic | $\eta \log Q^{\pi_p}$      | TD(1) | $\pi_p =\mathcal{N}(\mu_{\theta}(s), \Sigma_{\theta}(s))$ |
> > > | **RWR** | *EM* | Analytic | SG | $\eta \log r$ | -- | $\pi_p =\mathcal{N}(\mu_{\theta}(s), \Sigma)$  |
> > > | **REPS** | *EM* | Analytic | $\pi_q$ | $A^{\pi_p}$ | TD(0) | $\pi_p =$ Softmax |
> > > | **UREX** | *EM* | Analytic | SG | $Q^{\pi_p}$ | TD(1) | $\pi_p =$ Softmax |
> > > | **V-MPO** | *EM* | Analytic + TR | SG + TR | $A^{\pi_p}$ | $n$-step TD | $\pi_p =\mathcal{N}(\mu_{\theta}(s), \Sigma_{\theta}(s))$ |
> > > | **TRPO** | *KL* | TR | $\pi_q$ | $A^{\pi_p}$ | TD(1) | $\pi_q = \mathcal{N}(\mu_{\theta}(s), \Sigma_{\theta})$  |
> > > | **PPO** | *KL* | SG + TR | $\pi_q$ | $A^{\pi_p}$ | GAE | $\pi_q = \mathcal{N}(\mu_{\theta}(s), \Sigma_{\theta})$ |
> > > | **DDPG** | *KL* | SG | (Fixed) | $Q^{\pi_q}$ | TD(0) | $\pi_q = \mu_{\theta}(s)$ |
> > > | **TD3** | *KL* | SG | (Fixed) | $Q^{\pi_q}$ | TD(0) + TD3 | $\pi_q = \mu_{\theta}(s)$ |

---

> > > > ### Comment · Reviewer_34Q6 · 2021-08-20
> > > > **Comments on the empirical studies**
> > > >
> > > > Thanks for the updated table -- that is much clearer. One small note: it might be helpful to define what $\pi_\theta$ is parametrizing in the last column. It was defined in your original table but not in the new one.
> > > >
> > > > As for the empirical studies, my concerns and comments are as follows:
> > > > 1. Apart from the tanh squashing, you don't really look at any of the implementational details in Table 1. I think these would be of greater interest because they seem more central to these algorithms being control-as-inference algorithms. How transferable are the way the distributions get parametrized or how $G$ is estimated or the type of $G$ used. Instead, the details chosen seem less central to the control-as-inference framing.
> > > >
> > > > 2. Another question is whether there can be co-dependencies between the effectiveness of your implementational or code choices? For instance, perhaps you found choice A is less effective because it was paired with choice B, but it might be more effective if it was paired with choice B' instead.
> > > >     * in particular, you only considered one instance of a KL algorithm, SAC. But this algorithm also stands out because it is the sole algorithm using soft-Q. How do we know that the conclusions regarding SAC (e.g. regarding the effectiveness of clipped double Q-learning and the tanh-gaussianity) isn't because we're using a KL algorithm per se, but rather because we're using a soft-Q? In that case, your conclusions about clipped double Q-learning and the tanh-gaussianity wouldn't hold for the KL family in general.
> > > >     * the advice on network size is another example of this. We see differences in preferred network size depending on whether we use MPO or AWR. Yet both are EM algorithms. At the end of the day, isn’t the difference between them just different combinations of implementational/code choices? Then it stands to reason that the effect in network size you're seeing is because AWR used one particular combination of implementational/code choices while MPO used another, and the effect of network size is dependent on those choices rather than on whether the algorithm is from the EM or KL family.
> > > >     * more of an observation, but given the huge combinatorial space of possible implementational/code detail combinations, you might expect there to be several "local minima" in that space where specific combinations work well and changing any one code or implementational detail would break performance.
> > > >
> > > > 3. Related to point 2, your recommendations can read more like specific advice for particular implementational combinations rather than general statements about KL vs EM algorithms as a family. Again, one example is the recommendation that if you are going to use MPO (a specific combination of implementational/code choices), then you should use large networks. My concern is that these sort of recommendations are extremely specific and lack generality to be of sufficient utility to the community.
> > > >
> > > > These are my primary concerns about the empirical side of the paper.
> > > >
> > > > On a smaller note, it may also be helpful to include some summary statistics for each ablation study -- for instance, what fraction of tasks did we see an improvement by using X rather than Y.

---

> > > > > ### Author Response · Authors · 2021-08-20
> > > > > **Update of the table and plan of the experiments**
> > > > >
> > > > > We thank you for your quick response to our update. As you pointed out, we have updated the last column of Table 1 again, implying the parameterization choice of the policy ($\pi_p$ or $\pi_q$). We note that such choices are dependent on the algorithmic type; i.e. basically EM-control uses $\pi_p$ and KL-controls uses $\pi_q$ as the parameterized policy.
> > > > >
> > > > >
> > > > > **Concern 1:**
> > > > > We plan to include the additional comparison in (1) $\pi_p$ update and (2) $\mathcal{G}$.
> > > > > As for (1), we prepare MPO, AWR, AWAC with uniform-mixtured $\pi_p$ update; i.e. we also uses the actions sampled from uniform distribution as well as the samples from $\pi_{\theta_p^{k-1}}$ (as described in L158) for the M-step in EM-controls ($a_j \sim \alpha \text{Unif.} + (1-\alpha) \pi_{\theta_p^{k-1}}, \alpha \in (0, 1]$ ). These variants are much closer to SAC (using uniform distribution as $\pi_p$).
> > > > > In addition, we also prepare SAC with Tanh-Gaussian $\pi_p$; i.e. we copy the parameter of $\pi_q$ at a certain interval and use it as $\pi_p$ (L166). So, it seems “KL-regularized” actor-critic, rather than “soft” (entropy-regularized). This choice is much closer to MPO or AWR (AWAC).
> > > > >
> > > > > As for (2), we use soft-Q function (as in L214) for MPO and AWAC and regular Q function for SAC in the additional experiments. We note that we have partly addressed the choice of $\mathcal{G}$ (and $\mathcal{G}$ estimate) by the comparison between AWR and AWAC ($Q^{\pi_p}$-TD(0) vs $A^{\pi_p}$-TD($\lambda$); Figure 1).
> > > > > The results show that AWR is better at Hopper, Walker2dr and Swimmer, AWAC is better at Ant and HalfCheetah. Both of the performances seem equal in Humanoid. These results also imply the choice of  $\mathcal{G}$ and $\mathcal{G}$ estimate may not be so critical between the EM-control methods, which seems consistent with our observations in the author response: the robustness of EM-methods to implementation choices.
> > > > >
> > > > >
> > > > > **Concern 2:**
> > > > > We also plan to include experiments that directly address the co-dependent nature. As the reviewer suggested, we’ll test soft-Q MPO, AWAC, KL-SAC (as discussed above) with Tanh policy or clipped double Q-learning ablations. Also, we try to reveal the relationships between $\pi_p$ updates and the network size, using AWAC or MPO without trust-region for the comparison.
> > > > >
> > > > > **Concern3:**
> > > > > We’ll improve the recommendation statements towards more general ones through the additional experiments. In contrast to the network size example, we have already implied some of "general" advices. For example, both EM and KL control methods may not enjoy the benefit of clipped double Q-learning in non-terminal environments such as HalfCheetah or Swimmer. In addition, EM-controls should surely clip the actions to use tanh-gaussian policy.
> > > > >
> > > > >
> > > > > We'll post the additional results and insights from some of the experiments we listed above as soon as possible. We'll be glad if you take them into consideration for your decision.

---

> > > > > > ### Author Response · Authors · 2021-09-02
> > > > > > **Update the experimental results**
> > > > > >
> > > > > > We share the additional experiments, as posted above, to reveal deeper insights about the implementation details. We hope our attempt may address your concerns.

---

> > > > > > > ### Comment · Reviewer_34Q6 · 2021-09-03
> > > > > > > **Updated review**
> > > > > > >
> > > > > > > I thank the authors for their hard work on this. I've updated my review and score in light of the clarifications and new results, and I refer the authors there for my full comments.

---

> > > > > > > > ### Author Response · Authors · 2021-09-03
> > > > > > > > **Response to updated comment**
> > > > > > > >
> > > > > > > > We thank the reviewer for your update. As the reviewer pointed out, we agree that there are still a lot of margins to validate the co-dependent properties (maybe exponential combinations). We will continually address these problems current deep RL research faces, and hope that our works could build the foundations to encourage such directions.

---

> > > ### Comment · Area_Chair_jsSF · 2021-08-24
> > > **"survey" paper on RL algorithms**
> > >
> > > Well, in my years of reviewing and Area Chairing at NeurIPS, I cannot recall seeing a submission that, in essence, is a "survey" paper covering various RL algorithms and their relations.  I'm glad the authors took on this task, as it looks like it was a big effort to write it all up. By publishing at NeurIPS, I expect that many RL researchers will benefit by seeing different perspectives amongst the many algorithms, and could accelerate further advances over current RL methods.

---

> > > > ### Author Response · Authors · 2021-09-02
> > > > **Reply to the comment from AC**
> > > >
> > > > We appreciate the AC for a comment. As reviewer BvPF also mentioned, our "survey"-style paper seems few at NeurIPS. However, we believe that our unified framework and related EM-KL taxonomy could serve the analysis of existing or possible future algorithms. We will confidently emphasize the contribution of the classification between EM Control and KL Control, and of that between implementation details in the final revision.

---

### Official Review · Reviewer_CYmr · 2021-07-16

**Rating:** 6
**Confidence:** 3

**Summary:**

Provides a unifying mathematical framework for off-policy inference based algorithms such as MPO, AWR (that use pseudo-likelihood objective) and SAC, classifying them into EM based or KL based algorithms (uses the control as inference framework to do so). This allows decoupling the effect of high-level algorithmic choices from low-level code optimizations, to identify where the benefit from these algorithms really comes from, and further analyzing which design choices are transferrable across different algorithm families.

**Limitations And Societal Impact:**

The main limitation are novelty of the formulation (which has been used in prior work, albeit separately for the different algorithms), and the uncertainty over whether the low level design choice recommendations made here (with regard to network size, normalization etc) will apply to other agents/robots.


**Main Review:**

**Originality**: The control as inference framework has been used by the authors of MPO and SAC in their respective papers, and isn't a new way of looking at these algorithms. More specifically, the EM class formulation closely follows a similar characterization in the MPO paper. Nevertheless presenting these together as part of the same framework seems new (where KL control is shown to be almost a sub-problem for EM control). The paper cites a lot of the previous work dealing with meta-analysis of RL algorithms which point out that performance is often dependent on the specific code implementation used.

**Quality**: The mathematical formulation seems technically sound and well supported (and are based on prior work in the field). The experimental evaluation includes 5 openAI gym mujoco environments and is quite exhaustive as it includes ablations over a number of different design choices, such as network size, activations and normalization etc, in order to study which low-level code optimizations are important.

**Clarity**:  The paper is well written and easy to follow and understand. The authors first present their unified control as inference framework, then clearly show how MPO, AWR, SAC fall into it. The experimental evaluation is well organized for analysis into each separate ablation axis.

**Significance**: Irreproducibility is a well known issue for RL, especially the dependence of algorithm performance on the specific implementation used. This work takes a step towards addressing this issue. The exhaustive experimental evaluation might be useful to other researchers, where the authors analyze what aspects of the different algorithms are critical. They find that certain choices that are important for SAC (clipped double Q and squashed tanh), do not improve performance of other algorithms. Further they also find useful design choices in MPO (ELU and LayerNorm) which improve performance of SAC. That said, some of these observations might be specific to the gym environments tested, and may not transfer.

**Time Spent Reviewing:**

2

---

> ### Author Response · Authors · 2021-08-10
> **Author Response**
>
> We appreciate the reviewer for the careful reading of the paper and detailed feedback.
>
> **> Transferability of the experimental observations:**
> In addition to 6 MuJoCo tasks, we provided DM Control results over 28 tasks (Appendix C), and ablation of ELU and layer normalization in 12 of those tasks where MPO outperformed SAC (​​Figure 2 and Table 10 in Appendix C). In contrast, previous works of a large-scale empirical study evaluate design choices on 3 to 5 MuJoCo tasks (Andrychowics et al. (2021), Engstrom et al. (2020)). We believe the insights from our extensive evaluation are valuable for fairness and broadness, and can be transferable into other kinds of environments, such as DM Control.
>
> **> The novelty of the formulation:**
> As the reviewer pointed out, while the formulation follows the previous control-as-inference literature (Levine (2020), Haarnoja et al. (2018), Abdolmaleki et al. (2018)), we show that our unified objective and algorithmic taxonomy can cover most of the existing actor-critic algorithms (Table 1). This taxonomy could provide novel guidelines to the algorithm design (often, the novel algorithms might be proposed from different contexts and have large gaps in each other) and might help the researchers to look up the algorithmic property and to understand the algorithms that will be proposed in the future.
>
> ---
> **Reference**
>
> Andrychowicz et al. (2021). What matters for on-policy deep actor-critic methods? a large-scale study. In International Conference on Learning Representations.
>
> Engstrom et al. (2020). Implementation matters in deep rl: A case study on ppo and trpo. In International conference on learning representations.
>
> Levine (2018). Reinforcement learning and control as probabilistic inference: Tutorial and review. arXiv preprint arXiv:1805.00909.
>
> Haarnoja et al. (2018). Soft actor-critic: Off-policy maximum entropy deep reinforcement learning with a stochastic actor. In International conference on machine learning.
>
> Abdolmaleki et al. (2018). Maximum a posteriori policy optimisation.  In International conference on learning representations.

---

### Official Review · Reviewer_BvPF · 2021-07-18

**Rating:** 6
**Confidence:** 5

**Summary:**

This paper fits in the category of survey/review paper. While this kind of paper is not common at Neurips, it doesn't mean that it should be disregarded. It does a good job at trying to unify a popular body of work on policy gradient methods under the umbrella of "control as inference". The paper is well-structured: it defines a taxonomy which differentiates between "EM-based" methods and those based on "direct KL minimization". The paper focuses on four algorithms: Maximum a posteriori Policy Optimisation (Abdolmaleki et al. 2018), Advantage-Weighted Regression (Peng et al. 2019), Soft Actor-Critic (Haarnoja et al. 2017) and Relative Entropy Policy Search (Peters et al. 2010). The  paper starts by explaining how the RL problem can be cast as a probabilistic inference problem and tries to explain how an evidence lower bound can obtained. It then goes into the specifics of each algorithm and how each either parametrizes the prior or the posterior and whether  it uses hard constraints (with closed-form expressions) or soft/regularized re-formulations.

The empirical section then compares the raw performance three of those algorithms (SAC, MPO and AWR) on six Mujoco environments.

Regarding the "co-adaptation" expression in the title, I wouldn't say it defines the paper. The "co-adaptation" findings really pertain to the impact of specific implementation details on the final performance in those six Mujoco environments. More specifically, the authors study the impact of double-q learning + clipping, ELU activations vs ReLu, network size and layer normalization. The empirical findings are nicely summarized through tables.

**Limitations And Societal Impact:**

I don't want to be that reviewer who says: "more experiments". Of course we all want more experiments... But I need to be

I need to say that I'm becoming increasingly concerned about the over-reliance on the Mujoco domains for benchmarking policy gradient (Berkeley-style) methods in RL. As a community, we're using these environments so often that it's becoming increasingly difficult to assess where we stand in terms of "real-world" impact.  Are we overfitting?

Because this is essentially a survey paper, I would have like to see a more extensive evaluation.

How about taking more time and writing a much longer a journal paper? Like those great survey papers from the people at Darmstadt. How about the benchmark track at Neurips?

**Main Review:**

Overall, the structure is great and I like the ablatation study. Furthermore, I have learned new things via the EM vs direct minimization taxonomy. That being said, I think that the writing could be improved. I find the technical presentation is not solid enough. Not that the authors should go with a measure-theoretic presentation, but the paper would gain clarity if the notation is improved. Don't get me wrong, I'm not complaining about the choice of symbol, but rather the symbol-to-meaning mapping which I find deficient. For the kind of paper that the authors are aiming for, the background section needs to be near-perfect and crystal clear.

Regarding the main contribution, the co-adaptation claim, I think that it should be de-emphasized. Just in terms of space in the text, pages 1-6 are dedicated to the presentation of SAC, MPO and AWR under a unified narrative. The co-adapation findings are only developped in pages 7-9 and the overall takeaway seems to be that pretty much everything should stay the same for MPO and AWR but SAC is a bit more tolerant and can do better with a few tweaks taken from the other methods.

# Specific Comments

The expression "co-adaptation" is interesting, and I can see how it expresses the idea that you are trying to convey. That being said, I wonder if there would be another way to describe the issue without introducing new terminology (in this case borrowing from biology). For example, from the very begining in the abstract, the sentence:

"tanh Gaussian policy and network sizes are highly adapted to algorithmic types " (line 17) is a bit obscure to the non-initated

On line 53, "are highly co-adapted to KL-minimization-based", you could perhaps say "are highly specific" instead.

If you insists on using "co-adaptation", you should perhaps state explicitely what it means in the first sentence or so. Something like "we define co-adaptation as the implementation decisions that do not stem directly from the conceptual algorithmic development, but from empirical considerations."

On line 27, "there have been a series" I think that you should write "has"
On line 29, "on maximum entropy", "on a"

On line 88, "Our experiments prove", I suggest to keep "prove" on the math side. Instead: "demonstrate"/"highlight"

Caption of figure 1, "when the state-value function is parameterized instead of Q-function" ("instead of the"). This sentence fragment is unclear.

On line 93, "transition probability distribution" => "transition probability function", also "initial state distribution" not formally (Puterman) part of the MDP definition

Line 95: The use of "u" as a time index is unusual. How about "k" instead?

Line 95: "standard RL setting", I wouldn't say that this is necessary "standard". How about "we consider a policy gradient setting"?

Line 95: " for its parameter that maximizes" => "for parameters that maximize"

Line 99, " advantage function" cite Baird 1993.

Line 100, "is known to be a" => "is the"

Line 105, "For simplicity, we consider a finite-horizon MDP" I know that it is common in RL to make this assumption, but it's very important to keep in mind that the properties of finite-horizon MDPs and infinite-horizon discounted ones are not the same. The beauty of infinite horizon discounted is that it allows us to restrict our attention to stationary policies when searching for optimal policies. But it doesn't have to be the case in the finite horizon setting! In fact, the contraction/fixed-point based perspective is no longer appropriate. Instead, you would want to talk about backward induction and so on...

In the ELBO derivation around line 112, you have $\tau$ on the rhs but not the lhs. This is because $\mathcal{O}$ is function of $\tau$. Perhaps writing $\mathcal{O}(\tau)$ would make things clearer. Also, writing $\text{Pr}(\mathcal{O}|\pi_p)$ suggests that $\pi_p$ is a random variable, and that there is conditional distribution defined over those quantities. If so, I suggest being a bit clearer about that.

Line 116, "a popular choice is the one that factorizes", making that assumption of an MDP earlier, I don't think that you have much of a choice here in your factorization.

Line 123-124: are these other choices valid? As in: integrating/summing up to 1?

Line 125: You jump from an expectation over trajectories, to one where there exists a joint over state-action pairs. This is not a trivial step.

Line 132: "unnormalized state distribution", if it's unnormalized, it can't be a distribution and writing $\mathbb{E}_{d_\pi}$ doesn't quite make sense. think what you mean is something like $d_\pi = p_1^\top(I - \gamma P_\pi)^{-1}$. You then write an equality-constrained program where your equality constraints is the actual normalization. This is confusing.

Line 152: "converts the hard-constraint optimization". Just to be clear, the Lagrangian step is not about "converting" an constrained optimization problem into an unconstrained one (the constrained solution need not coincide with an unconstrained min or maximum of the Lagrangian).

Line 170: "an exact case". You should be clearer about what you mean by "exact" (lack of function approximation).

Line 173: "control problem, except the policy parameterization." The sentence sounds incomplete. Please expand

Line 182: "reverse KL penalty (second term) in the Lagrangian (Eq. 2) with a reverse KL constraint". This sentence is confusing, but I think what you mean is that it's actually solving a different constraint optimization problem. One where you not only require to have distributions (integrates to 1) but also that the KL is equal (or perhaps) less than some designated value. You should then write the equivalent constrained program as you did in equation 1. It would be clearer than talking about changes to the Lagrangian.

Line 194: "For a paramterized policy" => "parameterized" (you should use Grammarly to catch those prior to submitting)

Line 204: "replaced by samples of actions from a replay buffer" I don't quite understand how this works out to be similar.

Line 219: "the dual function of entropy constrained Lagrangian " I don't think that this sentence makes sense. I think all you want to say is that they treat $\eta$ as a Lagrange multiplier and they they then do gradient-ascent descent on the Lagrangian.

Section 5.1: "Distribution of the Policy". This is about the policy parametrization. Not about some "distribution over policies". I think that you should avoid this expression

**Time Spent Reviewing:**

3

---

> ### Author Response · Authors · 2021-08-10
> **Author Response**
>
> We thank the reviewer for the careful reading of the paper and constructive comments in detail.
>
>
> **> Definition of co-adaptation:**
> As the reviewer mentioned, we define “co-adaptation” as the indispensable choices in the implementation or code, that are not required from the algorithmic formulation. We also thank your comments on the abstract for the improvement. We will revise them properly.
>
>
> **> Response to Specific Comments:**
> We gratefully acknowledge your pointing out the wording issues in our manuscript. Our  comments are summarized as below:
>
> - Line 27, 88, 93, 95, 100, 170, 173, 194: we will replace them with the correct ones for the revision.
> - Caption of figure 1: we mean that we denote the algorithms that explicitly parameterize the $V(s)$ to compute advantage function, as $A^{\pi}$. In contrast, we denote the algorithms that use the Q-function, as $Q^{\pi}$. We rewrite it to avoid confusion.
> - Line 95, 105, 112, 125: we will clarify those notations and the context for the derivation properly.
> - Line 123-124: same as AWR, we can interpret these algorithms ignoring the normalization term $Z(s)$, and setting it to 1.
> - Line 116: we may use state-independent action distribution $p(a)$, instead of the state-dependent policy. In addition, we may consider the hierarchical settings introducing temporal abstracted actions.
> - Section 5.1: we will replace "Distribution of the Policy" with "Action Distribution for the Policy".
> - Line 99: we will add Baird 1993 to the reference.
> - Line 204: the stored actions in the replay buffer were sampled at the previous step. So when we sampled these actions uniformly, we can interpret them as the samples from the average policy over the previous iterations.
> - Line 132: we will replace “unnormalized state distribution” into “discounted state visitation distribution”
> - Line 152, 182, 219: As the reviewer pointed out, the current descriptions of the lagrangian seem a bit complicated. We will make them clearer and add an extra section in the Appendix for a more detailed description with some equations.
>
>
> **> Limitations: experimental coverage:**
> Our experiments cover possible choices in both implementation (clipped double Q-learning, Tanh-Gaussian) and code details (network size, activation function, normalization). We believe our careful ablation provides detailed insight for each algorithm (SAC, MPO, AWR).
> We’d like to know more concretely what kind of comparison isn’t sufficient for now. Could the reviewer let us know the ablation or evaluation that you think are important or should be included (e.g. another implementation details, or finer grid search of the current experiments such as network size, etc)?  We have an extensive codebase, and would be happy to address any specific additional ablations that you will suggest for the final version. We might deal with some of them and update the results by the end of the discussion period, the start of September.
>
>
> **> Limitations: environments for benchmark:**
> Our work follows the previous contexts, using 6 MuJoCo environments (Haarnoja et al. (2018)) and 28 DM Control environments (Abdolmaleki et al. (2018)), which is a quite diverse coverage compared to previous benchmark papers (3 to 5 MuJoCo tasks (Andrychowics et al. (2021) and Engstrom et al. (2020)). We’ll take it into consideration to include the other benchmark environments towards real-world applications, if those have been agreed to by the RL communities.
>
> ---
> **Reference**
>
> Andrychowicz et al. (2021). What matters for on-policy deep actor-critic methods? a large-scale study. In International Conference on Learning Representations.
>
> Engstrom et al. (2020). Implementation matters in deep rl: A case study on ppo and trpo. In International conference on learning representations.
>
> Levine (2018). Reinforcement learning and control as probabilistic inference: Tutorial and review. arXiv preprint arXiv:1805.00909.
> Haarnoja et al. (2018). Soft actor-critic: Off-policy maximum entropy deep reinforcement learning with a stochastic actor. In International conference on machine learning.
>
> Abdolmaleki et al. (2018). Maximum a posteriori policy optimisation.  In International conference on learning representations.

---

### Author Response · Authors · 2021-08-10
**Summary of Author Response to All the Reviewers and AC**

We would like to appreciate all the reviewers for their insightful comments. We will update the manuscript based on the constructive feedback from the reviewers. Our key responses to each reviewer are summarized as below:



### **[Clarification of implementation details]**

We thank the reviewer **34Q6** for pointing out the ambiguity of implementation details. We will explicitly separate current **“implementation details”** into **“implementation and code details”** in the revision, where new “implementation details” includes the components listed up in Table 1 ($\mathcal{G}$ estimate: clipped double Q-learning, $\pi_{\theta}$: Tanh-Gaussian Policy, in the experiments), and “code details” includes the details of neural network (network size, activation function, normalization), that is also a very important aspect of modern deep RL research as the previous empirical paper suggests (Andrychowicz et al. (2021), Engstrom et al. (2020), Henderson et al. (2018), Islam et al. (2017)).

Our extensive experimental observation suggests not only the co-adaptive nature and transferability of each implementation and code detail (discussed in Section 6), but also the properties of each kind of algorithm (KL-based and EM-based); **KL-based methods, such as SAC, shows the co-dependent nature in implementation details (clipped double Q-learning, Tanh-Gaussian Policy) but robustness to the code details related to neural networks. In contrast, EM-based methods, such as MPO and AWR, show the co-dependent nature in code details but robustness to the implementation details.** We believe these empirical observations from our experiments are valuable contributions to the RL community.


### **[Experimental Coverage]**

The reviewer **BvPF** and **34Q6** concerned about the experimental coverage. We’d like to clarify that our work focuses on not proposing best-performed algorithms or specifications, but revealing the effect of each implementation or code and algorithmic properties that each family of algorithm has, which has been often overlooked and less discussed in the algorithmic papers. Our unified formulation in Section 3 & 4 allows us such implementation and code comparison, and we successfully reveal the nature of both KL-based and EM-based algorithms, discussed in Section 5 & 6.
In addition, we believe that our experiments cover possible choices in both implementation (clipped double Q-learning, Tanh-Gaussian) and code details (network size, activation function, normalization), and that our careful ablation provides detailed insight for each algorithm (SAC, MPO, AWR).

**We would like to also ask the reviewer BvPF and 34Q6 for suggesting specific extra ablations they think are important.** We have an extensive codebase, and would be happy to address any specific additional ablations that you will suggest for the final version.
We might deal with some of them and update the results by the end of the rolling discussion period, the start of September.


### **[Contribution]**

As reviewer **BvPF** mentioned, the type of our paper, not proposing novel algorithms but analyzing and comparing the existing popular algorithms in detail, might be few at NeurIPS. However, we believe that our unified formulation and extensive comparison from both implementation and code perspective provides many contributions to the communities:

1. Our unified formulation can summarize and classify the recent inference-based off-policy actor-critic methods characterizing 7 components (2 from algorithms and 5 from implementations). We also show the applicability of our framework to the previous 9 algorithms (PoWER, RWR, REPS, UREX, V-MPO, TRPO, PPO, DDPG, TD3), proposed from independent literature. Such wider coverage of our work suggests the algorithms that might be proposed in the future could be classified following our taxonomy. We believe our formulation is an important foundation to interpret deep RL algorithms.

2. We experiment in both implementation and code details to reveal the transferability of each detail and algorithmic properties both KL-based and EM-based methods have. The algorithmic properties that each family of algorithms has has been often overlooked and less discussed in the algorithmic papers. We evaluate the choice of $\mathcal{G}$ estimate, action distribution for the policy (implementation details), network size, activation function, normalization (code details).
3. We use 6 MuJoCo environments and 28 DM Control tasks for the experiments, which is much more diverse than previous meta-analyses of RL papers (only 3 to 5 MuJoCo environments in Andrychowicz et al. (2021) and Engstrom et al. (2020)). This wider coverage of environments could guarantee the generalization or transferability to the other sort of environments. In addition, we run all the experiments with 10 random seeds.
4. Our empirical observation reveals that the co-adaptive nature between algorithm and implementation (e.g. clipped double Q-learning and Tanh-Gaussian Policy to SAC) and transferable choice (e.g. ELU and layer normalization to SAC or AWR and larger network size to SAC). Moreover, we reveal that KL-based methods, such as SAC, show the robustness to the code details, while EM-based methods, such as MPO and AWR, show the co-dependent nature in code details but robustness to the implementation details. Overcoming such drawbacks, implementation fragilities in KL-based methods and code fragilities in EM-based methods, might be important future directions to propose much better algorithms.

We hope our work can highlight the significance of considering the connection between algorithm formulations, and its implementation and code details, and can encourage more works to come out that study empirical design choices not only in one type of algorithms, but universally across a broader scope of deep RL algorithms.

---
**Reference**

Andrychowicz et al. (2021). What matters for on-policy deep actor-critic methods? a large-scale study. In International Conference on Learning Representations.

Engstrom et al. (2020). Implementation matters in deep rl: A case study on ppo and trpo. In International conference on learning representations.

Henderson et al. (2018). Deep reinforcement learning that matters. In Thirty-second AAAI conference on artificial intelligence.

Islam et al. (2017). Reproducibility of benchmarked deep reinforcement learning tasks for continuous control. arXiv preprint arXiv:1708.04133.

---

> ### Author Response · Authors · 2021-09-02
> **Additional experiments for deeper analysis of implementation details**
>
> We share the additional experimental results for deeper analysis of implementation details and co-adaptation nature, as reviewer 34Q6 suggested. We report the final cumulative return after 3M steps for Ant/HalfCheetah/Walker2d/Swimmer, 1M steps for Hopper, and 10M steps for Humanoid. All results below are averaged among 10 random seeds.
>
> We will reflect these results appropriately in the main text (following EM-KL comparison as we discussed before) and include the rest in the Appendix. We also hope all of you take these results into consideration for your decision.
>
> **(1) $\pi_p$ Update**
>
> We test different types of $\pi_p$ Update to investigate the effectiveness of implementation choices. We prepare 4 variants: (i) MPO with a uniform prior, (ii) AWAC with a uniform prior, (iii) SAC with target policy instead of a fixed uniform prior, (iv) MPO without trust-region (only SG update). The details of (i)-(iv) are described below:
>
> (i) & (ii): we use the actions sampled from uniform distribution as well as the samples from $\pi_{\theta_p^{k-1}}$ (as described in L158) for the M-step in EM-controls ($a_j \sim \alpha Unif. + (1-\alpha) \pi_{\theta_p^{k-1}}, \alpha \in (0, 1]$ ). These variants are much closer to SAC (using uniform distribution as $\pi_p$). We test $\alpha=\{0.25, 0.5, 0.75\}$ for both MPO and AWAC.
>
> (iii): we copy the parameter of $\pi_q$ at a certain interval and use it as $\pi_p$ (L166), similar to MPO/PPO/TRPO. It seems “KL-regularized” actor-critic, rather than “soft” (entropy-regularized). We test both constant (\{0.1, 0.01, 0.001\}) and lagrangian coefficients (KL target=\{1.0, 0.1, 0.01 \}).
>
> (iv): original MPO stabilize the $\pi_p$ update incorporating TR (trust-region) into SG. We test the effect of TR, just removing TR term in the M-step of MPO.
>
> However, the variants listed above have shown drastic degradation compared to the original choice (we omit the table since most of them failed). For example, the larger $\alpha$ in (i) & (ii) we chose, the lower scores the algorithm achieved. Also, KL-SAC didn’t learn meaningful behaviors. These failures suggest that the implementation choice of $\pi_p$ Update might be the most important one and should be designed carefully for both KL and EM controls.
>
>
> **(2) $\mathcal{G}$: Soft Q function**
>
> We investigate the effect of the soft Q function, instead of standard Q function as MPO or AWAC use. We prepare (i) MPO with soft Q, (ii) AWAC with soft Q, and (iii) SAC without soft Q, just modifying Bellman equation and keep the policy objectives as they are.
>
> The results show that SAC without soft Q degrades its performance over 5 tasks except for Ant, but it is not so drastic compared to clipped double Q or Tanh-Gaussian policy.
> In contrast, MPO with soft Q slightly improves the performance (over 4 tasks), and AWAC with soft Q slightly also does (over 3 tasks). These trends are similar to the clipped double Q or Tanh-Gaussian policy. We think these experiments also support our empirical observation: KL-based methods, such as SAC, show the robustness to the code details, while EM-based methods, such as MPO and AWR, show the co-dependent nature in code details but robustness to the implementation details.
>
> | *Method* | *Hopper-v2* | *Walker2d-v2* | *HalfCheetah-v2* | *Ant-v2* | *Humanoid-v2* | *Swimmer-v2* |
> | -------- | -------: | -------: | -------: | -------: | -------: | -------: |
> | **SAC**| $3013\pm602$ | $5820\pm411$ | $15254\pm751$ | $5532\pm1266$ | $8081\pm1149$ | $114\pm21$ |
> | **SAC (w/o Soft Q)**| $2487\pm870$ | $5674\pm202$ | $12319\pm2731$ | $6496\pm305$ | $6772\pm3060$ | $114\pm33$ |
> | **MPO** | $2136\pm1047$ | $3972\pm849$ | $11769\pm321$ | $6584\pm455$ | $5709\pm1081$ | $70\pm40$ |
> | **MPO (Soft Q)** | $2271\pm1267$ | $3817\pm794$ | $11911\pm274$ | $6312\pm332$ | $6571\pm461$| $80\pm32$ |
> | **AWAC** | $2329\pm1020$ | $3307\pm780$ | $7396\pm677$ | $3659\pm523$ | $5243\pm200$ | $35\pm8$ |
> | **AWAC (Soft Q)**| $2545\pm1062$ | $3671\pm575$ | $7199\pm628$ | $3862\pm483$ | $5152\pm162$ | $35\pm10$ |
>
> **(3) Network size for AWAC**
>
> To investigate the co-dependent nature between implementation and code details, we add the network size ablation of AWAC, whose implementations stand between MPO and AWR.
> AWAC differs $\pi_p$ update and network size (the default choice of AWAC is (M)) from MPO (MPO uses TD(0) in open-source implementation & the difference of $\pi_{\theta}$ might be minor). Also, AWAC differs $\mathcal{G}$ and $\mathcal{G}$ estimate from AWR.
>
> The results of AWAC show a similar trend to AWR in high-dimensional tasks (Ant, Humanoid); a larger network didn’t help. We may hypothesize that $\pi_p$ update of AWR/AWAC, mixture+SG, is not good at optimizing larger networks, compared to SG + TR of MPO. In contrast, especially, Hopper and Walker2d show a similar trend to MPO; the larger, the better. Totally, AWAC with different network sizes shows the mixture trend of AWR and MPO, which is the same as implementation details.
> We think these observations highlight the co-adaptation nature between implementation and code details.
>
> | *Method* | *Hopper-v2* | *Walker2d-v2* | *HalfCheetah-v2* | *Ant-v2* | *Humanoid-v2* | *Swimmer-v2* |
> | -------- | -------: | -------: | -------: | -------: | -------: | -------: |
> | **AWAC (L)** | $2769\pm919$ | $4350\pm542$ | $6433\pm832$ | $2342\pm269$ | $4164\pm1707$ | $40\pm5$ |
> | **AWAC (M)** | $2329\pm1020$ | $3307\pm780$ | $7396\pm677$ | $3659\pm523$ | $5243\pm200$ | $35\pm8$ |
> | **AWAC (S)** | $2038\pm1152$ | $2022\pm971$ | $5864\pm768$ | $3705\pm659$ | $5331\pm125$ | $34\pm11$ |
>
> **(4) Clipped double Q learning/Tanh-Gaussian + Soft Q function**
>
> Both clipped double Q learning/Tanh-Gaussian policy and soft Q function are the important implementation choices to KL control, SAC, which lead to significant performance gains. To test the co-adaptation nature more in detail, we implement these two choices into MPO and AWAC.
>
> The results show that incorporating such multiple choices doesn’t show any notable improvement in EM controls, MPO and AWAC. They support the co-adaptation nature of those two implementations to SAC.
>
> | *Method* | *Hopper-v2* | *Walker2d-v2* | *HalfCheetah-v2* | *Ant-v2* | *Humanoid-v2* | *Swimmer-v2* |
> | -------- | -------: | -------: | -------: | -------: | -------: | -------: |
> | **MPO (S)** | $2136\pm1047$ | $3972\pm849$ | $11769\pm321$ | $6584\pm455$ | $5709\pm1081$ | $70\pm40$ |
> | **MPO (D)** | $2352\pm959$ | $4471\pm281$ | $12028\pm191$ | $7179\pm190$ | $6858\pm373$ | $69\pm29$ |
> | **MPO (Soft Q, S)** | $2271\pm1267$ | $3817\pm794$ | $11911\pm274$ | $6312\pm332$ | $6571\pm461$| $80\pm32$ |
> | **MPO (Soft Q, D)** | $1283\pm632$ | $4378\pm252$ | $12117\pm126$ | $6822\pm94$ | $6895\pm433$ | $45\pm4$ |
> | **AWAC (S)** | $2540\pm755$ | $3662\pm712$ | $7226\pm449$ | $3008\pm375$ | $2738\pm982$ | $38\pm7$ |
> | **AWAC (D)** | $2329\pm1020$ | $3307\pm780$ | $7396\pm677$ | $3659\pm523$ | $5243\pm200$ | $35\pm8$ |
> | **AWAC (Soft Q, S)**| $2732\pm660$ | $3656\pm416$ | $7270\pm185$ | $3494\pm330$ | $2926\pm1134$ | $36\pm10$ |
> | **AWAC (Soft Q, D)**| $2545\pm1062$ | $3671\pm575$ | $7199\pm628$ | $3862\pm483$ | $5152\pm162$ | $35\pm10$ |
>
> | *Method* | *Hopper-v2* | *Walker2d-v2* | *HalfCheetah-v2* | *Ant-v2* | *Humanoid-v2* | *Swimmer-v2* |
> | -------- | -------: | -------: | -------: | -------: | -------: | -------: |
> | **MPO** | $2136\pm1047$ | $3972\pm849$ | $11769\pm321$ | $6584\pm455$ | $5709\pm1081$ | $70\pm40$ |
> | **MPO (Soft Q)** | $2271\pm1267$ | $3817\pm794$ | $11911\pm274$ | $6312\pm332$ | $6571\pm461$| $80\pm32$ |
> | **MPO (Soft Q, Tanh)** | $314\pm8$* | $368\pm47$* | $3427\pm207$* | $628\pm221$* | $5919\pm202$* | $35\pm8$* |
> | **AWAC** | $2329\pm1020$ | $3307\pm780$ | $7396\pm677$ | $3659\pm523$ | $5243\pm200$ | $35\pm8$ |
> | **AWAC (Soft Q)**| $2545\pm1062$ | $3671\pm575$ | $7199\pm628$ | $3862\pm483$ | $5152\pm162$ | $35\pm10$ |
> | **AWAC (Soft Q, Tanh)** | $2989\pm484$ | $2794\pm1692$ | $6263\pm247$ | $3507\pm458$ | $66\pm4$ | $32\pm5$ |
>
>
> (* numerical error happens during training)

---

### Decision · Program_Chairs · 2021-09-27

**Decision:**

Accept (Poster)

**Comment:**

While the reviewers thought there were various ways that this paper could be improved, there was also general consensus that this framework which unites multiple existing algorithms was interesting and useful, prompting one reviewer to comment on the new insights it brought them. The authors are encouraged to address the feedback from the reviewers in their camera ready.